# SOURCE-GUIDED FLOW MATCHING

**Zifan Wang**[*]
KTH Royal Institute of Technology
zifanw@kth.se

**Alice Harting**[*]
KTH Royal Institute of Technology
aharting@kth.se

**Matthieu Barreau**
KTH Royal Institute of Technology
barreau@kth.se

**Michael M. Zavlanos**
Duke University
michael.zavlanos@duke.edu

**Karl H. Johansson**
KTH Royal Institute of Technology
kallej@kth.se

## ABSTRACT

Guidance of generative models is typically achieved by modifying the probability flow vector field through the addition of a guidance field. In this paper, we instead propose the Source-Guided Flow Matching (SGFM) framework, which modifies the source distribution directly while keeping the pre-trained vector field intact. This reduces the guidance problem to a well-defined problem of sampling from the source distribution. We theoretically show that SGFM recovers the desired target distribution exactly. Furthermore, we provide bounds on the Wasserstein error for the generated distribution when using an approximate sampler of the source distribution and an approximate vector field. The key benefit of our approach is that it allows the user to flexibly choose the sampling method depending on their specific problem. To illustrate this, we systematically compare different sampling methods and discuss conditions for asymptotically exact guidance. Moreover, our framework integrates well with optimal flow matching models since the straight transport map generated by the vector field is preserved. Experimental results on synthetic 2D benchmarks, physics-informed generative tasks, and imaging inverse problems demonstrate the effectiveness and flexibility of the proposed framework.

## 1 INTRODUCTION

Flow matching (Lipman et al., 2022) is a generative modeling framework to learn a vector field that drives the probability flow from a source distribution $q_0$ to a target distribution $q_1$ in some fixed time. It has demonstrated state-of-the-art computational efficiency and sample quality across a range of applications, from image generation (Lipman et al., 2022) and molecular structure generation (Chen and Lipman, 2023) to decision-making tasks (Zheng et al., 2023). In particular, optimal flow matching (Tong et al., 2023) trains the vector field by leveraging the optimal transport (OT) solution between $q_0$ and $q_1$. The resulting optimal vector field moves each sample along a straight-line trajectory with a constant velocity, corresponding to the Wasserstein geodesic between the distributions. In practice, these straight trajectories lead to stable training and faster inference, since generative sampling can then reach the target distribution with few integration steps.

The guidance of flow matching refers to directing the probability flows toward outcomes with desired properties (Venkatraman et al., 2025; Dhariwal and Nichol, 2021; Du et al., 2023; Graikos et al., 2022; Ho and Salimans, 2022; Song et al., 2020). In this context, sample generation aims not only to approximate the data distribution but also to satisfy additional properties, such as conditioning on auxiliary information or optimizing an energy-based objective. Training-based guidance methods (Ho and Salimans, 2022; Song et al., 2020) address this by training a specialized model for

---

[*]First two authors have equal contribution.

a given conditioning scenario. While effective, these methods require retraining for every new conditioning scenario, which incurs significant cost and therefore limits their flexibility. Thus, a variety of training-free approaches have emerged for both diffusion models (Chung et al., 2022; Song et al., 2023; Ye et al., 2024; Uehara et al., 2024; Tang, 2024) and flow matching models (Ben-Hamu et al., 2024; Wang et al., 2024; Liu et al., 2023; Domingo-Enrich et al., 2024; Feng et al., 2025).

Among these methods, exact guidance is achieved by Uehara et al. (2024); Tang (2024); Domingo-Enrich et al. (2024); Feng et al. (2025). Specifically, Uehara et al. (2024); Tang (2024) reformulate guidance as a stochastic optimal control (SOC) problem and achieve exactness by modifying both the source distribution and the vector field. Additionally, Domingo-Enrich et al. (2024) shows that exact guidance is possible by modifying only the vector field, given a suitable noise schedule. However, these methods require solving an SOC problem for every new conditioning scenario, which is computationally expensive. Recently, Feng et al. (2025) proposed a framework for exact guidance including various adjustments of the vector field. However, exactness is only proved for a specific class of pre-trained vector fields, thereby having limited generality. Moreover, a shared feature of all of these methods is that the vector field is transformed. For optimal flow matching models, this means that the desirable property of straight-line transport is not preserved when guidance is applied.

In this work, we show that exact guidance can instead be achieved by appropriately modifying the source distribution while keeping the original vector field unchanged. We introduce the Source-Guided Flow Matching (SGFM) framework, which reduces guidance to the well-defined task of sampling from the modified source distribution. We prove that sampling from the modified source distribution and driving the flow along the exact vector field precisely recovers the desired target distribution. Furthermore, we provide bounds on the Wasserstein error of the target distribution when using an approximate sampler of the source distribution and an approximate vector field.

The key to effective implementation of SGFM is accurate and efficient sampling of the modified source distribution. SGFM gives the user the flexibility to tune the procedure by customizing the sampling method according to their specific problem. Such methods include importance sampling, Hamiltonian Monte Carlo, and optimization-based sampling. We discuss the asymptotic exactness of SGFM with these methods. Interestingly, SGFM with optimization-based sampling method coincides with the heuristic formulation in Ben-Hamu et al. (2024). In the context of our framework, this method is equivalent to recovering the mode of the modified source distribution. In this way, we offer a new view of Ben-Hamu et al. (2024) with theoretical justification that also naturally extends to other sampling methods. Experiments on synthetic 2D datasets, physics-informed generative tasks, and imaging inverse problems demonstrate the effectiveness and flexibility that SGFM offers compared to other methods.

## 2 BACKGROUND

Throughout this paper, we consider a generative modeling framework defined on a data space in $\mathbb{R}^d$. The generative model is characterized by a source distribution $q_0$ and a target distribution $q_1$. The source distribution is an arbitrary distribution from which samples can be drawn, while the target distribution represents an empirical data distribution given by a finite set of samples.

### 2.1 PROBABILITY FLOW AND FLOW MATCHING

The goal of a flow-based generative model is to sample from the target distribution $q_1$ by transforming samples from the source distribution $q_0$. Specifically, the model is defined by a vector field $u_t(x) : [0, 1] \times \mathbb{R}^d \to \mathbb{R}^d$, which transports particles according to the ordinary differential equation (ODE) $dx = u_t(x)dt$. Integration yields the transport map $\phi_t(x_0)$, which maps the initial point $x_0$ to the solution $x_t$ at time $t$. Applying $\phi_t$ to a distribution of particles $p_0$ induces a probability flow where the density at time $t$ given by the pushforward measure $p_t \triangleq [\phi_t]_\#(p_0)$, where $[g]_\#$ is defined by the property $\int f(x) \, d([g]_\#(p))(x) = \int f \circ g(x) \, dp(x)$ with $f \circ g(x) \triangleq f(g(x))$ for every integrable function $f$ (Figalli and Glaudo, 2021). Equivalently, $p_t$ can be characterized as the probability flow arising from the continuity equation $\partial_t p_t + \nabla \cdot (p_t u_t) = 0$ (Villani et al., 2008).

In this view, the flow matching problem is to find a vector field $u_t$ that induces a probability flow $p_t$ such that $p_0 = q_0$ and $p_1 = q_1$. While an exact vector field $u_t$ is often inaccessible, it can be

approximated by a neural network $v_t^\theta$ and trained using the conditional flow matching objective

$$\mathcal{L}_{\text{FM}}(\theta) = \mathbb{E}_{t \in \mathcal{U}[0,1],(x_0,x_1) \sim \pi} \left\| v_t^\theta((1-t)x_0 + tx_1) - (x_1 - x_0) \right\|^2, \tag{1}$$

where the joint distribution $\pi \in \Gamma(q_0, q_1)$, with $\Gamma(q_0, q_1)$ being the set of all joint distributions with marginals $q_0$ and $q_1$ (Lipman et al., 2022). For example, we can select $\pi(x_0, x_1) = q_0(x_0) \times q_1(x_1)$.

## 2.2 Optimal transport and optimal flow matching

There are many possible transport plans between $q_0$ and $q_1$; among these, the optimal transport (OT) plan is defined as the minimizer of the total cost of transportation. This is quantified by the 2-Wasserstein distance, which is expressed in Kantorovich or Monge formulations respectively as

$$W_2^2(q_0, q_1) = \min_{\pi \in \Gamma(q_0,q_1)} \int_{\mathbb{R}^d \times \mathbb{R}^d} \|x - y\|^2 \, \mathrm{d}\pi(x,y) = \min_{T: T_\# q_0 = q_1} \int_{\mathbb{R}^d} \|x - T(x)\|^2 \, q_0(x) \, \mathrm{d}x.$$

As shown in Villani (2021); Figalli and Glaudo (2021), these optimization problems admit unique minimizers $\pi^*$ and $T^*$, which are related by $\pi^* = [\text{Id}, T^*]_\# q_0$. Of particular interest to our case is the dynamic OT formulation, which is defined by the optimization problem:

$$W_2^2(q_0, q_1) = \inf_{(p_t, u_t)} \left\{ \int_0^1 \int_{\mathbb{R}^d} \|u_t(x)\|^2 \, p_t(x) \, dx \, dt \, \middle| \, \begin{array}{l} \partial_t p_t + \nabla \cdot (p_t u_t) = 0, \\ p_0 = q_0, \ p_1 = q_1 \end{array} \right\}, \tag{2}$$

which seeks the vector field $u_t^*$ that induces a probability flow $p_t$ that transports the source distribution $p_0 = q_0$ to the target distribution $p_1 = q_1$ with minimal total kinetic energy. The relation between the static and dynamic OT solutions is simply given as $u_t^*((1-t)x_0 + tT^*(x_0)) = T^*(x_0) - x_0$. Thus, $u_t^*$ gives rise to a linear trajectory $x_t = tT^*(x_0) + (1-t)x_0$ for every initial point $x_0$.

Among the infinitely many choices of vector fields that solve the flow-matching problem, the unique solution $u_t^*$ to the dynamic OT formulation in equation 2 is associated with particularly efficient inference and fast generation. This is because $u_t^*$ is independent of $t$, so ODE integration along this field simply yields straight-line paths, which lead to lower time-discretization errors and improved computational efficiency (Kornilov et al., 2024; Liu et al., 2022). To approximate $u_t^*$ via equation 1, it is necessary to choose $\pi = \pi^*$. However, computing $\pi^*$ has cubic computational complexity in the number of samples, which is challenging for large datasets. A solution is to instead approximate $\pi^*$ using mini-batch data (Tong et al., 2023), or alternatively to use entropic OT solvers (Pooladian et al., 2023). Another approach is to train flow matching models on a class of vector fields that guarantee straight trajectories (Kornilov et al., 2024).

## 2.3 Flow matching guidance

Given a pre-trained flow matching model that transforms the source distribution $q_0$ to the target distribution $q_1$, consider the conditional generation problem where the task is to generate samples that satisfy additional constraints. When the constraints are encoded by an energy function $J$, which attains its minimum when the constraints are satisfied, the likelihood of constraint satisfaction can be expressed in canonical form $\propto e^{-J(\cdot)}$. In this case, the new target distribution becomes $q_1'(x_1) \propto q_1(x_1) \times e^{-J(x_1)}$. It can be shown that $q_1'$ is the solution of the variational problem $q_1' = \arg\min_q \mathbb{E}_{x_1 \sim q}[J(x_1)] + \text{KL}(q\|q_1)$, where $\text{KL}(q\|q_1)$ denotes the Kullback-Leibler divergence between $q$ and $q_1$ (Uehara et al., 2024). In this view, conditional generation is a fine-tuning problem: the distribution is shifted to reduce the task-specific loss $J$ while staying close to the original data distribution $q_1$.

# 3 Source-Guided Flow Matching

Suppose that we have a pre-trained flow matching model $v_t$ that transports the source distribution $q_0$ to the target distribution $q_1$. Consider the conditional generation task in which the new target distribution is of the form $q_1'(x_1) \propto q_1(x_1) \times e^{-J(x_1)}$, where $J$ is a given loss function. The problem considered in this paper is how to generate samples from $q_1'$. To that end, one could, in principle, modify the source distribution and/or the vector field. Here, we explore how to to generate a flow that arrives at $q_1'$ by modifying only the source distribution while retaining the vector field.

### 3.1 EXACT GUIDANCE UNDER AN EXACT TRANSPORTATION MAP

Consider the ideal case where the pre-trained vector field $v_t$ exactly transports $q_0$ to $q_1$. In this case, we derive a closed-form expression for the modified source distribution. We show that transporting samples from the modified source distribution along $v_t$ precisely yields the desired target distribution. This result is formally stated in the following theorem and proven in Appendix A.1.

**Theorem 1.** *Let $q_0$ and $q_1$ be the source and target distributions, respectively. Let $v_t \colon \mathbb{R}^d \to \mathbb{R}^d$ be a vector field whose flow map $\phi_t$ satisfies $(\phi_1)_{\#} q_0 = q_1$. For any measurable function $J \colon \mathbb{R}^d \to \mathbb{R}$, define the new target distribution $q_1'(x_1) = \frac{1}{Z_1} q_1(x_1) e^{-J(x_1)}$ and new source distribution $q_0'(x_0) = \frac{1}{Z_0} q_0(x_0) e^{-J \circ T(x_0)}$, where $T = \phi_1$, and $Z_0, Z_1$ are normalizing constants. Then, the same flow $\phi_t$ transports $q_0'$ to $q_1'$, i.e., $(\phi_1)_{\#} q_0' = q_1'$.*

Theorem 1 indicates that, if $x_0 \sim q_0'$ and $x_t$ evolves as $dx_t = v_t(x_t)dt$, then $x_1 = T(x_0) \sim q_1'$. In other words, exact guidance is achieved. Inspired by this theorem, we propose the SGFM framework, presented in Algorithm 1. First, we learn a vector field $v_t^\theta$ by minimizing the flow matching loss in equation 1. Then, we draw samples $x_0 \sim q_0'$, using an appropriate sampling strategy such as those discussed Section 4. Finally, each sample $x_0$ is transported along the learned vector field $v_t^\theta$ by integrating the associated ODE, yielding guided samples $x_1$.

---

**Algorithm 1:** Source-Guided Flow Matching

1: **Input**: Source samples $x_0 \sim q_0$, target data samples $x_1 \sim q_1$, loss function $J$
2: Train the vector field $v_t^\theta(\cdot)$ that transforms $q_0$ to $q_1$
3: Sample $x_0 \sim q_0'$
4: Integrate over ODE $\frac{d}{dt} x_t = v_t^\theta(x_t)$
5: **Output**: samples $x_1$

---

### 3.2 ERROR ANALYSIS UNDER VECTOR FIELD AND SOURCE SAMPLING APPROXIMATIONS

Learning an exact vector field is inherently difficult, particularly in high-dimensional spaces. In addition, exact sampling from the desired source distribution $q_0'$ may not be possible. In this section, we analyze how these errors jointly influence the quality of the generated samples. Specifically, we quantify how deviations in both the vector field and the source distribution contribute to the divergence between the target and generated distributions.

To that end, let $v_t(x)$ denote the exact vector field, and let $v_t^\theta(x)$ denote the learned vector field with flow $\phi_t^\theta(x)$, such that $\frac{d}{dt} \phi_t^\theta(x) = v_t(\phi_t^\theta(x))$. We derive an upper bound on the error of the conditionally generated distribution in terms of the two error sources; the proof is given in Appendix A.2.

**Theorem 2.** *Assume that $\left\| v_t(x) - v_t^\theta(x) \right\|_\infty \leq \epsilon$, and the learned flow $v_t^\theta(x)$ is $L_v$-Lipschitz continuous in $x$. Suppose that the sampling method returns samples of distribution $\tilde{q}_0$. Then, the generated samples of distribution $[\phi_1^\theta]_{\#} \tilde{q}_0$ satisfy $W_2(q_1', [\phi_1^\theta]_{\#} \tilde{q}_0) \leq e^{L_v} W_2(q_0', \tilde{q}_0) + \epsilon e^{L_v}$.*

To interpret Theorem 2, the first term reflects the distributional discrepancy introduced by an approximate sampler of the source distribution scaled by $e^{L_v}$, the Lipschitz constant of the flow map $\phi_1^\theta$. Intuitively, any deviation in the initial distribution can be amplified by at most a factor of this Lipschitz constant during transport. The second term captures the accumulated effect of errors in the learned vector field over the trajectory. This contribution arises from integrating a bounded drift perturbation $\epsilon$ over an $L_v$-Lipschitz flow, where $L_v$ controls local error growth along trajectories. Thus, $L_v$ characterizes the sensitivity of the generative process to errors in both source distribution and vector field. When the vector field is perfectly learned ($\epsilon = 0$), exact guidance is feasible. From Theorem 2, we conclude that our guidance method is particularly effective when $L_v$ is small.

### 3.3 IMPROVED GUIDANCE WITH THE OPTIMAL VECTOR FIELD

To encourage a small Lipschitz constant $L_v$, we employ methods designed to learn the optimal vector field $v_t^*$ (Tong et al., 2023); see Table 1 for more details and experimental support. The optimal vector field $v_t^*$ also improves efficiency, as it induces straight trajectories with constant velocity, thereby reducing the number of discretization steps needed to integrate the ODE over $t \in [0, 1]$ to obtain the flow map $T = \phi_1$. In addition to accelerating particle transportation, this also lowers the cost of sampling from the modified source distribution $q_0'(x_0) = \frac{1}{Z_0} q_0(x_0) e^{-J \circ T(x_0)}$, since this

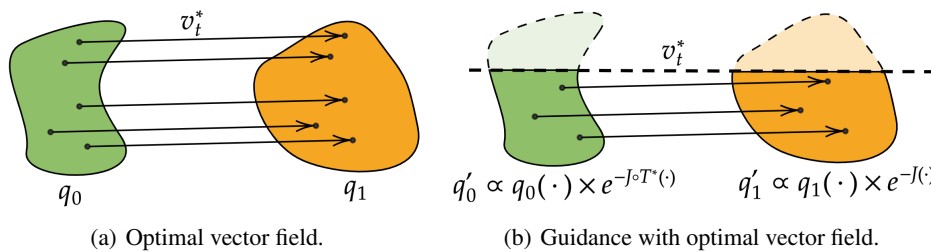

(a) Optimal vector field.                (b) Guidance with optimal vector field.

Figure 1: Illustration of SGFM

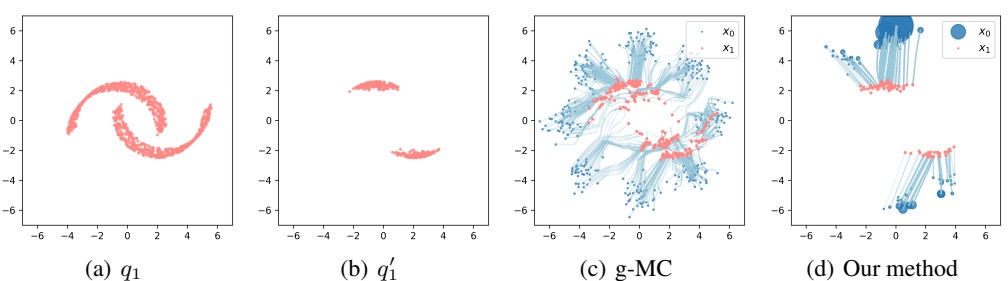

(a) $q_1$                (b) $q_1'$                (c) g-MC                (d) Our method

Figure 2: Illustration of flow guidance in a 2D example. The method g-MC (Feng et al., 2025) modifies the vector field, resulting in curved trajectories. In contrast, our proposed method maintains the optimal vector field and modifies the source distribution, leading to straight trajectories. In (d), each ball's radius indicates the relative weight of the corresponding source sample.

requires evaluating the flow map $T$ by definition. Theoretically, the flow map under $v_t^*$ coincides with the optimal Monge map $T^*$ (Figalli and Glaudo, 2021, Theorem 4.1.3).

Our method is illustrated in Figure 1. In the prior model, $v_t^*$ induces a straight-line map from each source sample $x_0 \sim q_0$ to its corresponding target sample $x_1 \sim q_1$. Sampling from the conditional target $q_1'$ by minimizing $J$ reduces to selecting the subset of the source samples that flows to $q_1'$ under $v_t^*$. Theorem 1 shows that this subset follows the distribution $q_0'(x_0) \propto q_0(x_0)e^{-J \circ T^*(x_0)}$, where $T^*$ is the flow map at $t = 1$. Because our guidance method modifies only the source distribution, the straightness of $v_t^*$ is preserved in conditional generation, ensuring inference speed and integration stability. By contrast, guidance methods modifying the vector field (e.g., Feng et al. (2025)) yield curved trajectories, as illustrated in Figure 2, requiring finer discretization to maintain accuracy.

**Modifying the source distribution in $\mathbb{R}^d$ versus modifying the vector field in $\mathbb{R}^d \times [0, 1]$:** In conditional generation, several existing methods (Song et al., 2023; Feng et al., 2025) augment the original vector field $v_t$ with a guidance term $g_t$, typically approximated via Monte Carlo sampling. Since generation involves evaluating the augmented vector field at numerous intermediate times $t \in [0, 1]$, where each evaluation demands many samples, the whole process requires extensive sampling. In contrast, our approach leaves the vector field unchanged, and transforms the task into sampling from a modified source distribution at a single time. This is enabled by an instantiation of the classical change-of-variables formula (see also Venkatraman et al. (2025)), as formalized in Theorem 1. Our framework naturally decomposes the error into just two distinct sources, whose scaling factors directly motivate using optimal transport-based models for superior accuracy (Theorem 2).

## 4 SAMPLING FROM THE MODIFIED SOURCE DISTRIBUTION

Given the pre-trained optimal vector field $v_t^*$, the guidance problem is reduced to drawing samples from the modified source distribution. Thus, the key to effective implementation of our method is accurate and efficient sampling from $q_0'(x_0) \propto q_0(x_0)e^{-J \circ T^*(x_0)}$. The choice of the sampling method depends on the properties of the cost function $J$ and the dimensionality of the sample space. Whenever the sampling method generates a sequence of approximate distributions $(\tilde{q}^k)_{k \geq 0}$ such that $W_2^2(\tilde{q}^k, q_0') \to 0$ as $k \to \infty$, our method of guided flow matching is asymptotically exact, as

follows from Theorem 2. In this section, we discuss asymptotically exact samplers and efficient, optimization-based approximations, including their connection to D-Flow (Ben-Hamu et al., 2024). Additional relevant sampling methods are described in Appendix B.2.

### 4.1 ASYMPTOTICALLY EXACT SAMPLING METHODS

**Importance Sampling:** In low-dimensional spaces, importance sampling (IS) (Chopin and Papaspiliopoulos, 2020) offers a fast and gradient-free sampling method. Given an unnormalized target distribution $q$, an initial set of particles is generated using a proposal distribution $m$ such that $\text{supp}(q) \subset \text{supp}(m)$. Samples are then drawn from this set according to weights determined by their relative probability in the target versus proposal distribution $W^n = \frac{w(X^n)}{\sum_m w(X^m)}$, where $w(x) \propto \frac{q(x)}{m(x)}$.

The approximate distribution $\tilde{q}^N(x) \triangleq \sum_{n=1}^N W^n \delta_{X^n}(x)$, with $X^n \sim m$, converges weakly to the target distribution $q$ when $N \to \infty$ (Chopin and Papaspiliopoulos, 2020). Assuming $q$ is defined on a closed and bounded subset of $\mathbb{R}^d$, this implies that $W_2^2(\tilde{q}^N, q) \to 0$ as $N \to \infty$ (Villani, 2009, Theorem 6.9). When $q = q'_0$ and $m = q_0$, we have $w(x_0) = e^{-J \circ T^*(x_0)}$. Note that this method does not require $J$ to be differentiable. For a detailed outline of the algorithm, see Appendix B.2.1.

**Hamiltonian Monte Carlo:** IS suffers from the curse of dimensionality, making Hamiltonian Monte Carlo (HMC) (Neal et al., 2011) a popular alternative in high-dimensional state spaces. HMC is a gradient-based Markov chain Monte Carlo method for unnormalized, continuous densities. In particular, HMC generates proposal samples by propagating the target variable, representing position in space, and an auxiliary momentum variable using Hamiltonian dynamics, achieving extensive exploration while maintaining high acceptance probabilities. For more details, see Appendix B.2.3.

HMC typically returns an ergodic Markov chain, which means that it converges asymptotically to the target distribution $q$ (Neal et al., 2011). When the negative log-likelihood $-\ln q$ is twice differentiable, strongly convex and has Lipschitz-continuous gradients, and the integration of the dynamics is sufficiently accurate, the law of the Markov chain after $N$ steps $\tilde{q}^N$ approximates the target distribution $q$ up to arbitrarily small Wasserstein precision $W_2^2(\tilde{q}^N, q)$ for a sufficiently large $N$ (Chen and Vempala, 2019, Theorem 5).

In our case, the negative log likelihood is $-\ln q'_0(x_0) = -\ln q_0(x_0) + J \circ T^*(x_0)$, where $T^*$ generally prevents global convexity. However, since $T^*$ is learned while encouraging straight-line transport, we might expect that the composition $J \circ T^*$ approximately preserves the convexity of $J$ locally. To escape local modes, various strategies exist, e.g., tempering (Neal et al., 2011).

### 4.2 OPTIMIZATION-BASED SAMPLING

While HMC offers strong theoretical guarantees, its computational cost can become prohibitive for highly non-concave target densities. An approximate and more efficient alternative is to dispense with the momentum variable and acceptance step, and instead directly search for high-probability regions of the target distribution through the optimization problem:

$$\min_{x_0} -\ln q'_0(x_0) \quad \Leftrightarrow \quad \min_{x_0} -\ln q_0(x_0) + J \circ T^*(x_0). \tag{3}$$

In this formulation, the term $J \circ T^*(x_0)$ introduces task-specific loss via the OT map $T^*$, while the term $-\ln q_0(x_0)$ acts as a regularizer. This regularizer, however, attracts $x_0$ toward the most probable fixed points, rather than toward regions of high probability. For example, when the source distribution is Gaussian $x_0 \sim q_0 = \mathcal{N}(0, I_d)$, the optimization problem in equation 3 becomes

$$\min_{x_0} \|x_0\|^2 / 2 + c + J \circ T^*(x_0), \tag{4}$$

in which the regularizer $-\ln q_0(x_0) = \|x_0\|^2 / 2 + c$ would guide the sample toward the unique mode at $x_0 = 0$ and lead to mode collapse.

To mitigate this issue, the regularizer can be replaced by an alternative that better promotes sample diversity. For the Gaussian source distribution, we have $\|x_0\|^2 \sim \chi^2$, where $\chi_d^2$ is the chi-square distribution with $d$ degrees of freedom. Instead of regularizing with the probability of the sample, one might instead use the probability of the norm of the sample, $-\ln p_{\chi_d^2}(\|x_0\|^2)$. This means that

the unique prior mode at $x_0 = 0$ is replaced by the sphere $\{x_0 : \|x_0\|^2 = \arg\max_x p_{\chi_d^2}(x) = \max(d-2,0)\}$. The resulting optimization problem is

$$\min_{x_0} -\ln p_{\chi_d^2}(\|x_0\|^2) + J \circ T^*(x_0) \Leftrightarrow \min_{x_0} -(d-2)\log\|x_0\| + \|x_0\|^2/2 + J \circ T^*(x_0), \quad (5)$$

which coincides with heuristic formulations in Ben-Hamu et al. (2024) (see discussion below).

**Relation to D-Flow (Ben-Hamu et al., 2024):** In D-Flow, sampling is heuristically reformulated as an optimization problem with various forms of regularization, where representative instances coincide with equation 4 and equation 5. Since our formulation guarantees fidelity to the ground truth target distribution via Theorem 1, we explicitly clarify the role of the regularization term, thus providing foundational support. Therefore, D-Flow can be regarded as a special case of SGFM.

An extension of this idea is to more explicitly target high-density regions of the prior. Since $\mathbb{E}[\|x_0\|^2] = d$, $\mathrm{Var}[\|x_0\|^2] = 2d$, and $p_{\chi_d^2}$ is unimodal, we observe that the prior density concentrates in the hyperspherical shell $|\|x_0\|^2 - d| \leq \sqrt{2d}$. Motivated by this, we propose a new method within the optimization-based sampling family for Gaussian source distributions given by

$$\min_{x_0} J \circ T^*(x_0) \quad \text{s.t. } |\|x_0\|^2 - d| \leq \sqrt{2d}. \quad (6)$$

In practice, the constrained problem in equation 6 can be addressed either by incorporating a regularizer of the form $(\|x_0\|^2 - d)^2$, or by applying projected gradient descent onto the feasible set. The implementation details can be found in Appendix B.2.4.

From Theorem 1, it follows that ensuring that $x_0$ lies in high-density regions of $q_0'$ implies that the corresponding sample $x_1$ will also lie in high-density regions of $q_1'$, justifying equation 3 – equation 6 as sampling methods in our framework. In practice, we can design specific optimization objectives to sample from $q_0'$ depending on the problem structure and source distribution (Appendix B.2.5).

The optimization-based sampling method is typically suitable when the target distribution resembles a Dirac distribution, or when we are interested in obtaining a high-probability sample rather than a representative sample of the distribution. However, it risks leading to mode collapse: under perfect optimization, the global mode is consistently returned. In practice, some diversity among the samples can still be seen if the optimizer is randomly initialized and the objective function has multiple local maxima from which it struggles to escape. Then, the samples concentrate around these optima, resulting in a "local" mode collapse. We illustrate this with an example below.

**Example of mode collapse in optimization-based sampling:** Consider a set of particles at locations $(x^1, x^2)$ uniformly distributed over an 'S'-shaped structure. To encourage $x_1$ and $x_2$ to be close, we introduce a soft penalty $J = \|x^1 - x^2\|$ to guide generation. As shown in Figure 3, applying equation 3, or equivalently D-Flow, leads to an excessive concentration of the particles around the line $x_1 = x_2$. Therefore, optimization-based sampling fails to capture the inherent diversity of the true conditional distribution.

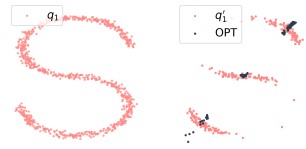

Figure 3: Mode collapse in optimization-based sampling

## 5 EXPERIMENTS

In this section, SGFM is evaluated on toy 2D examples, physics-informed generative tasks, and inverse imaging problems. Our framework is benchmarked against its closest counterparts: D-Flow (Ben-Hamu et al., 2024), top-performing methods in Feng et al. (2025), and PnP-flow (Martin et al., 2024).

### 5.1 TOY 2D EXAMPLE

We begin the evaluation on a two-dimensional synthetic dataset, consisting of a uniform source distribution with

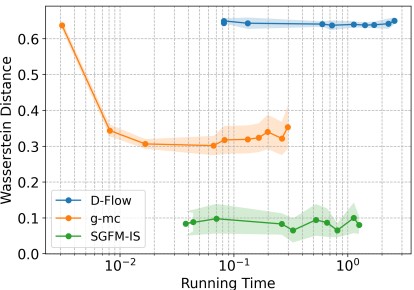

Figure 4: Comparison of guidance precision and running time in 2D example.

an 8-Gaussian target distribution. Since the source is non-Gaussian, diffusion-based guidance cannot be applied. We use importance sampling (SGFM-IS) and evaluate performance in terms of guidance precision, measured by the empirical Wasserstein distance between the true guided distribution and the generated distribution, relative to inference time, as controlled by the number of function evaluations (NFEs). As shown in Figure 4, our method consistently achieves better guidance precision. Moreover, reducing NFEs (which lowers runtime), has only a small effect on precision. This observation aligns with the prior findings that the optimal vector field produces straight trajectories that require fewer integration steps.

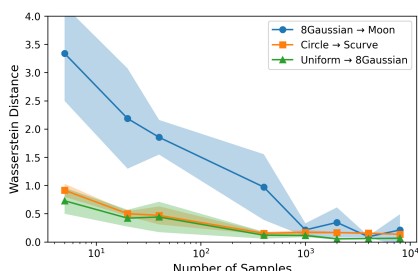

Figure 5: Asymptotic exactness

Next, we evaluate the exactness of SGFM, in terms of how guidance precision evolves as the number of IS samples increases. As shown in Figure 5, for each of three pairs of two-dimensional source and target distributions, the guidance precision consistently improves as the IS samples increase. Hence, this low-dimensional experiment shows that SGFM-IS achieves asymptotic exactness.

We evaluate the impact of pre-trained vector fields on guidance performance by training the vector fields in equation 1 using two different coupling strategies: an independent coupling $\pi = q_0 \otimes q_1$ and the optimal coupling $\pi^*$, referred to as the independent vector field and optimal vector field, respectively. The Lipschitz constant $L_v$ for each field is empirically approximated using autodifferentiation. Guidance performance is quantified using the Wasserstein distance between the true target distribution and the generated distribution. As shown in Table 1, the optimal vector field is associated with a smaller Lipschitz constant and achieves a lower guidance error. This result aligns with Theorem 2, confirming that a smaller Lipschitz constant is associated with reduced sensitivity and improves generation performance.

Table 1: Guidance performance and approximated Lipschitz parameters for two pre-trained vector fields. The optimal vector field consistently demonstrates superior performance with lower Lipschitz constants, consistent with Theorem 2.

|  | 8gaussian → moon | | uniform → 8gaussian | |
| --- | --- | --- | --- | --- |
|  | $L_v$ | Guidance error | $L_v$ | Guidance error |
| Independent vector field | 20.1 | $0.125 \pm 0.186$ | 16.8 | $0.124 \pm 0.023$ |
| Optimal vector field | 11.9 | $0.066 \pm 0.047$ | 11.1 | $0.067 \pm 0.019$ |

## 5.2 PDE INVERSE PROBLEM

We next consider a high-dimensional inverse problem with a multi-modal posterior distribution based on the Darcy flow equations (Bastek et al., 2024; Jacobsen et al., 2025). Darcy flow is an elliptic PDE describing fluid flow though a porous medium with permeability field $K$ and pressure field $p$. The flow matching model is trained to sample pairs of $K$ and $p$ occurring as discretized solutions on a square domain with resolution $64 \times 64$. The dataset (Bastek et al., 2024) is obtained by solving the PDE using finite differences. For more details, see Appendix C.2.1–C.2.2.

The conditional sampling problem is to generate permeability fields consistent with a partially observed pressure field. We define the family of valid solutions as the target distribution. The validity of an inverse estimate $\hat{K}$ is measured by $J(p_{\hat{K}})$, where $J$ computes the target reconstruction error and $p_{\hat{K}}$ is the true pressure field corresponding to $\hat{K}$. Since $p_{\hat{K}}$ is inaccessible, the sampling guidance cost is $J(\hat{p})$, where $\hat{p}$ is the pressure field sampled jointly with $\hat{K}$. We assess the performance of SGFM using HMC and two variants of optimization-based sampling, SGFM-OPT-1 (equation 4) and SGFM-OPT-2 (equation 5), the latter equivalent to Ben-Hamu et al. (2024) with preferred regularization, and compare with g-covA, g-covG (Feng et al., 2025) and PnP flow (Martin et al., 2024).

Figure 6 shows the target pressure and $p_{\hat{K}}$ corresponding to a single outcome of $\hat{K}$ for each method. SGFM-OPT-2 obtains the best target reconstruction, closely followed by SGFM-HMC. In comparison, SGFM-OPT-1 and baseline methods suffer from large biases. Additional samples are shown

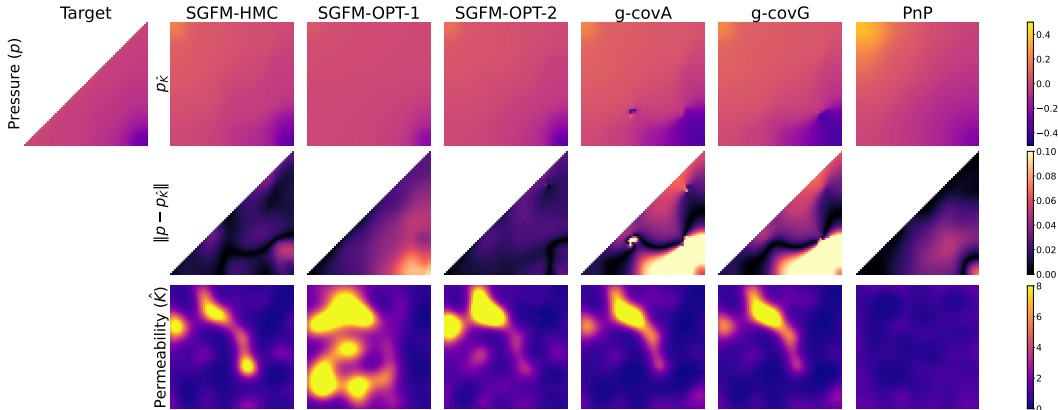

Figure 6: Solutions to the inverse problem of the Darcy flow equations. Top: target pressure and true solution $p_{\hat{K}}$ corresponding to inverse estimate $\hat{K}$; middle: target reconstruction error; bottom: inverse estimate $\hat{K}$ of the permeability field generated by conditional sampling.

in Appendix C.2.3. To further analyse the performance, the validity $J(p_{\hat{K}})$, guidance cost $J(\hat{p})$, and physical consistency $\|p_{\hat{K}} - \hat{p}\|$ of 25 samples are shown in Table 2, reported as the median and interquartile range. SGFM-OPT-2 achieves the best validity, followed by SGFM-HMC. In comparison, SGFM-OPT-1 and baseline methods do not perform significantly better than unconditional sampling. Although g-covA achieves the lowest guidance cost, it compromises physical consistency, leading to poor validity. Similarly, SGFM-OPT-1 has worse validity than SGFM-HMC despite achieving lower guidance cost. In contrast, both SGFM-HMC and SGFM-OPT-2 maintain physical consistency while SGFM-OPT-2 achieves lower guidance cost, resulting in the best validity. The poor performance of PnP can be attributed to its difficulty in handling the target's multi-modality: by interpolating a gradient step with random noise followed by denoising, this method jumps between different modes rather than converging consistently to a single one, resulting in degraded accuracy.

Due to the complex source distribution, the SGFM methods require longer runtimes compared to g-covA, which limits the number of samples that can be evaluated. While this constrains our ability to assess how well the methods capture the whole family of solutions, we show that SGFM-HMC performs best in this regard for an example in lower dimension in Appendix C.5.

Table 2: Performance of guidance methods in the Darcy flow inverse problem

| Method | Validity of Inverse Estimate ($\downarrow$) | Guidance Cost ($\downarrow$) | Physical Consistency ($\downarrow$) |
| --- | --- | --- | --- |
| SGFM-HMC | **0.591 [0.532, 0.654]** | 0.281 [0.248, 0.335] | **0.188 [0.168, 0.228]** |
| SGFM-OPT-1 | 0.907 [0.503, 1.875] | 0.206 [0.149, 0.294] | 0.421 [0.174, 0.770] |
| SGFM-OPT-2 | **0.474 [0.416, 0.562]** | 0.187 [0.131, 0.218] | **0.194 [0.157, 0.215]** |
| g-covA | 0.992 [0.857, 1.293] | **0.030 [0.028, 0.052]** | 0.289 [0.247, 0.351] |
| g-covG | 0.955 [0.814, 1.201] | 0.242 [0.163, 0.388] | 0.245 [0.190, 0.285] |
| PnP | 1.055 [0.950, 1.204] | 0.610 [0.590, 0.632] | **0.116 [0.099, 0.129]** |
| Unconditional sampling | 1.006 [0.860, 1.269] | 1.051 [0.905, 1.289] | 0.214 [0.167, 0.274] |

## 5.3 Imaging inverse problem on CelebA

We evaluate SGFM on various imaging inverse problems using the high-dimensional CelebA dataset ($\mathbb{R}^{3\times128\times128}$) (Yang et al., 2015), considering five distinct tasks: denoising, deblurring, super-resolution, random inpainting, and box inpainting. Since the target distribution for these inverse problems is typically Dirac or highly concentrated, we apply optimization-based sampling within our SGFM framework. We index the SGFM-OPT variants by 1-6, where OPT-1 corresponds to equation 4 and OPT-2 to equation 5 (Ben-Hamu et al. (2024)) as before, and OPT-(3–5), OPT-6 correspond to equation 6 implemented with different regularizers and projected gradient descent, respectively (Table 3). Our methods are benchmarked against strong baselines, including top methods

g-covA and g-covG from Feng et al. (2025), and PnP flow (Martin et al., 2024). For details of the SGFM variants, implementation, and visualizations of the generated images, see Appendix C.3.

Table 3: Regularizer or constraint for variants of optimization-based sampling.

| Method | Regularizer in $\min_{x_0} R(x_0) + J \circ T^*(x_0)$ or constraint |
|---|---|
| SGFM-OPT-1 equation 4 | $R_1(x_0) = \|x_0\|^2$ |
| SGFM-OPT-2 equation 5 | $R_2(x_0) = -\ln p_{\chi_d^2}(\|x_0\|^2) = -(d-2)\log\|x_0\| + \frac{\|x_0\|^2}{2}$ |
| SGFM-OPT-3 equation 6 | $R_3(x_0) = (\|x_0\|^2 - d)^2$ |
| SGFM-OPT-4 equation 6 | $R_4(x_0) = \big|\|x_0\|^2 - d\big|$ |
| SGFM-OPT-5 equation 6 | $R_5(x_0) = (\|x_0\| - \sqrt{d})^2$ |
| SGFM-OPT-6 equation 6 | Constraint: $\big|\|x_0\|^2 - d\big| \le \sqrt{2d}$ |

The results in Table 4 show that SGFM variants achieve state-of-the-art performance. Specifically, they outperform g-covA and g-covG in all tasks. Our method is competitive with PnP-flow in most tasks and ranks one class below in deblurring; however, we note that PnP is specifically designed for imaging inverse problems, while our method is more general. The results demonstrate that SGFM with optimization-based samplers is an effective and flexible method for imaging inverse problems.

Table 4: PSNR (↑) comparison of methods for inverse problems on CelebA.

| Method | Denoising | Deblurring | Super-res | Rand inpaint | Box inpainting |
|---|---|---|---|---|---|
| g-covA | 26.73 | 29.72 | 18.45 | 19.61 | 24.88 |
| g-covG | 30.35 | 29.50 | 24.18 | 25.49 | 26.12 |
| PnP | **32.14** | **38.74** | 31.33 | 33.87 | 29.92 |
| SGFM-OPT-1 | 28.51 | 35.12 | 33.30 | 34.02 | 28.51 |
| SGFM-OPT-2 (D-Flow) | 28.95 | 35.23 | **33.32** | 34.01 | 28.43 |
| SGFM-OPT-3 | 31.51 | 35.21 | 33.28 | **34.05** | **30.09** |
| SGFM-OPT-4 | **31.60** | **35.27** | 33.31 | 34.03 | **30.12** |
| SGFM-OPT-5 | 28.94 | 35.22 | **33.33** | **34.06** | 28.55 |
| SGFM-OPT-6 | 31.54 | 32.60 | 32.10 | 32.36 | 29.19 |

We evaluate the sensitivity of the method to the optimality of the vector field. Recall that, while our method requires no straightness assumptions (Theorems 1–2 hold for arbitrary vector fields), this is motivated by practical considerations: increased straightness enables lower NFEs and thus faster inference without sacrificing performance. In practice, mini-batch OT training (Tong et al., 2023) yields sufficiently straight vector fields to enable effective guidance: as shown in Table 5, increasing NFE beyond NFE = 3 has little to no benefit on SGFM performance in the CelebA experiment.

Table 5: PSNR (↑) SGFM-OPT-2 performance using increasing NFEs on CelebA

| NFE | Denoising | Deblurring | Super-res | Rand inpaint | Box inpainting |
|---|---|---|---|---|---|
| NFE = 1 | 21.33 | 34.71 | 32.65 | 31.57 | 29.83 |
| NFE = 3 | 28.64 | **35.38** | **32.95** | **32.92** | 29.61 |
| NFE = 6 | 29.16 | 34.53 | 32.59 | 32.72 | 30.21 |
| NFE = 9 | **29.31** | 34.43 | 32.30 | 32.69 | **30.57** |

## 6 CONCLUSION

We presented a framework for guided flow matching with theoretical guarantees. The framework reduces the guidance problem to a problem of sampling from a modified source distribution. Examples on 2D benchmarks, physics-informed generative tasks, and imaging inverse problems demonstrated the effectiveness and flexibility of the framework. We acknowledge that sampling from the source distribution may present its own challenges, especially for complex, high-dimensional distributions. Nevertheless, the proposed method offers users the flexibility to select a sampling strategy that balances their desired trade-offs between accuracy and computational cost.

ACKNOWLEDGMENTS

This work was supported in part by the Swedish Research Council Distinguished Professor Grant (2017-01078), a Knut and Alice Wallenberg Foundation Wallenberg Scholar Grant, Wallenberg AI, Autonomous Systems and Software Program (WASP) funded by the Knut and Alice Wallenberg Foundation, and Swedish Strategic Research Foundation SUCCESS Grant FUS21-0026. Computational resources were provided by the National Academic Infrastructure for Supercomputing in Sweden (NAISS) at C3SE, partially funded by the Swedish Research Council under Grant Agreement No. 2022-06725.

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

# A    PROOFS

## A.1    PROOF OF THEOREM 1

Recall that $T = \phi_1$. Since the transport map $T$ pushes $q_0$ to $q_1$, according to Villani (2009) we have

$$q_1(x_1) = T_\# q_0(x_1) = \frac{q_0(T^{-1}(x_1))}{|\det \nabla T^{-1}(x_1)|}. \tag{7}$$

The resulting pushforward distribution of $q_0'$ under the transport map $T$ is then

$$T_\# q_0'(x_1) = \frac{q_0'(T^{-1}(x_1))}{|\det \nabla T^{-1}(x_1)|} = \frac{q_0(T^{-1}(x_1))e^{-J(x_1)}}{Z|\det \nabla T^{-1}(x_1)|} = \frac{q_1(x_1)e^{-J(x_1)}}{Z} = q_1'(x_1), \tag{8}$$

where the second equality follows from the definition of $q_0'$ and third equality follows from equation 7. The proof is complete.

We note that the extension to to discrete state spaces,

$$q_1(x_1) = T_\# q_0(x_1) = \sum_{x_0 \in T^{-1}(x_1)} q_0(x_0),$$

is straightforward: By defining again $q_0'(x_0) := \frac{1}{Z_0} q_0(x_0) e^{-J \circ T(x_0)}$, the modified proof follows as

$$T_\# q_0'(x_1) = \sum_{x_0 \in T^{-1}(x_1)} q_0'(x_0) \propto \sum_{x_0 \in T^{-1}(x_1)} q_0(x_0) e^{-J \circ T(x_0)} \propto q_1(x_1) e^{-J(x_1)} \propto q_1'(x_1).$$

## A.2    PROOF OF THEOREM 2

Before the proof of Theorem 2, we give a useful lemma.

**Lemma 1.** *Suppose that $f(x)$ is $L$-Lipschitz continuous in $x$. Then, we have*

$$W_2(f_\# \mu, f_\# \nu) \leq L W_2(\mu, \nu). \tag{9}$$

*Proof.* The 2-Wasserstein definition gives

$$W_2^2(f_\# \mu, f_\# \nu) = \inf_{\pi' \in \Gamma(f_\# \mu, f_\# \nu)} \mathbb{E}_{(f(x), f(y)) \sim \pi'} \| f(x) - f(y) \|^2.$$

Denote $(f \times f)(x, y) = ((f(x), f(y)))$. For every $\pi' \in \Gamma(f_\# \mu, f_\# \nu)$, we have a corresponding $\pi \in \Gamma(\mu, \nu)$ that satisfies $\pi' = (f \times f)_\# \pi$. Then, we have

$$
\begin{aligned}
W_2^2(f_\# \mu, f_\# \nu) &= \inf_{\pi \in \Gamma(\mu, \nu)} \mathbb{E}_{(x, y) \sim \pi'} \| f(x) - f(y) \|^2 \\
&\leq L^2 \inf_{\pi \in \Gamma(\mu, \nu)} \mathbb{E}_{(x, y) \sim \pi'} \| x - y \|^2 \\
&= L^2 W_2^2(\mu, \nu),
\end{aligned}
\tag{10}
$$

where the inequality follows from the Lipschitz property of $f$. Taking the square root on both sides of equation 10 completes the proof. $\square$

Now we are ready to prove Theorem 2. By virtue of the triangle inequality for the Wasserstein distance (Santambrogio, 2015, Lemma 5.3), we have

$$W_2(q_1', [\phi_1^\theta]_\# \tilde{q}_0) = W_2([\phi_1]_\# q_0', [\phi_1^\theta]_\# \tilde{q}_0) \tag{11}$$

$$\leq W_2([\phi_1]_\# q_0', [\phi_1^\theta]_\# q_0') + W_2([\phi_1^\theta]_\# q_0', [\phi_1^\theta]_\# \tilde{q}_0). \tag{12}$$

Since the learned vector field has uniform error bound $\epsilon$ and is $L_v$-Lipschitz continuous, by virtue of Theorem 1 in Benton et al. (2023), the first term can be bounded by

$$W_2([\phi_1]_\# q_0', [\phi_1^\theta]_\# q_0') \leq \epsilon e^{L_v}. \tag{13}$$

In what follows, we analyze the Lipschitz property of $\phi_t^\theta$. Recall that $\phi_t^\theta$ is the flow of the learned vector field $v_t^\theta$. Let $x_t$ and $y_t$ be the solutions of the ODEs

$$dx_t = v_t^\theta(x_t)dt, \quad x_0 = x_0$$
$$dy_t = v_t^\theta(y_t)dt, \quad y_0 = y_0,$$

respectively. Define $\Delta_t = \|x_t - y_t\|^2$. Then, we have

$$\frac{d\Delta_t}{dt} = 2\langle x_t - y_t, \frac{dx_t}{dt} - \frac{dy_t}{dt}\rangle = 2\langle x_t - y_t, v_t^\theta(x_t) - v_t^\theta(y_t)\rangle \leq 2L_v \|x_t - y_t\|^2 = 2L_v\Delta_t.$$

Integrating from $0$ to $t$ gives

$$\Delta_t \leq \Delta_0 + 2L_v \int_0^t \Delta_s ds.$$

By virtue of Grönwall's inequality, we have

$$\Delta_t \leq \Delta_0 e^{2L_v t} = \|x_0 - y_0\|^2 e^{2L_v t}.$$

By taking the square root, we have that $\phi_t^\theta(x)$ is $e^{L_v t}$-Lipschitz continuous in $x$. In particular, at $t = 1$, $\phi_1^\theta(x)$ is $e^{L_v}$ Lipschitz continuous. Then, it follows from Lemma 1 that the second term is bounded by

$$W_2([\phi_1^\theta]_\# q_0', [\phi_1^\theta]_\# \tilde{q}_0) \leq e^{L_v} W_2(q_0', \tilde{q}_0). \tag{14}$$

Substituting equation 13 and equation 14 into equation 11, we have the desired result. The proof is complete.

## B ADDITIONAL DISCUSSIONS

### B.1 RELATED WORKS

**Diffusion guidance:** Conditional sampling has been widely studied in diffusion models (Chung et al., 2022; Song et al., 2023; Ye et al., 2024; Guo et al., 2024; Wu et al., 2023; Xu and Chi, 2024; Bruna and Han, 2024). However, the diffusion model requires the source distribution to be Gaussian, and cannot handle general source distributions. Therefore, these guidance methods cannot be applied here.

**Flow matching guidance:** The flow guidance methods can be divided into two groups: training-based guidance and training-free guidance. Training-based guidance (Zheng et al., 2023) requires retraining when we have a different conditioning. Therefore, this paper focuses on training-free guidance (Ben-Hamu et al., 2024; Feng et al., 2025). One closely related training-free guidance method is D-Flow (Ben-Hamu et al., 2024), which proposes to optimize the source samples via a regularized optimization problem. However, its optimization objective is heuristic, whereas our framework provides the missing theoretical foundation. Besides, Feng et al. (2025) proposed a training-free guidance method that keeps the original source distribution and modifies the vector field. Such an approach generates curved vector fields and, therefore, requires a large number of discretization steps to integrate the ODE. Moreover, the exactness of this guidance method applies to a limited class of pre-trained vector fields and lacks generality.

**Guidance via stochastic optimal control (SOC):** Optimal control methods have been used to guide generative models (Uehara et al., 2024; Tang, 2024; Wang et al., 2024; Domingo-Enrich et al., 2024). Specifically, Wang et al. (2024) augments the vector field with an additional control term, obtained by solving a SOC problem. However, Wang et al. (2024) does not connect the generated distribution with the target distribution, and there is a bias between these two distributions. The works Uehara et al. (2024); Tang (2024) cancel out this bias by both modifying the vector field and shifting the initial distribution. More recently, Domingo-Enrich et al. (2024) showed that solely adjusting the vector field is able to remove the bias if the noise schedule is appropriately selected. Our method is orthogonal to the guidance methodology of Domingo-Enrich et al. (2024): we remove the bias by solely shifting the source distribution. Moreover, whenever we have a new guidance energy function, SOC-based guidance methods have to re-solve the SOC problem, which is computationally expensive.

**Guidance by optimizing the source distribution:** Conditional generation by optimizing the source distribution has been explored in Ben-Hamu et al. (2024); Wallace et al. (2023); Tang et al. (2024); Novack et al. (2024); Karunratanakul et al. (2024). These works propose to propagate the gradient from the target criteria through the whole generation process to update the initial noise. However, their optimization objectives are heuristically designed without theoretical justification. One exception is Venkatraman et al. (2025), which provides a justification for achieving exact guidance by modifying only the source distribution. We extend this line of work by giving an error analysis that explicitly delineates when such approaches are effective. Moreover, whereas Venkatraman et al. (2025) emphasizes training-based guidance, our focus is on training-free methods.

### B.2 SAMPLING ALGORITHMS

A key strength of SGFM is that it reduces guidance to a well-defined sampling problem, enabling users to flexibly choose the most suitable method for their specific problem. For low-dimensional problems, importance sampling is a simple and asymptotically exact method, as described in Section 4.1. For more complex distributions where gradient information is available and a minor bias acceptable, the Unadjusted Langevin Algorithm (ULA) (Robert et al., 1999) offers an efficient option. If asymptotic exactness is required and the computational budget is larger, a better alternative is Metropolis–Adjusted Langevin Algorithm (MALA) (Robert et al., 1999), which corrects ULA's bias. For extensive exploration of complex, high-dimensional distributions, HMC would be preferred. In the most complex cases, we may use optimization-based approximate samplers as described in Section 4.2. Below, we give further details on some of these methods. Beyond classical methods, promising approaches to approximate sampling include variational inference and recent diffusion-based samplers (Berner et al., 2022), cf. Venkatraman et al. (2025). Implementation of these is left for future work.

#### B.2.1 IMPORTANCE SAMPLING

Following the discussion on importance sampling (IS) in Section 4.1, a detailed outline of the method with target density $q = q_0'$ and proposal density $m = q_0$ is given in Algorithm 2 (Chopin and Papaspiliopoulos, 2020).

---
**Algorithm 2:** Importance Sampling

1: **Input**: samples from $x_0 \sim q_0$
2: Set $w(\cdot) \triangleq \frac{q_0'(\cdot)}{q_0(\cdot)} = e^{-J \circ T^*(\cdot)}$
3: Compute weights $W^n = \frac{w(x_0^n)}{\sum_m w(x_0^m)}$
4: Sample $x_0'$ from $\{x_0^n\}$ with probabilities $\{W^n\}$
5: **Output**: sample $x_0'$ from $q_0'$

---

#### B.2.2 UNADJUSTED AND METROPOLIS ADJUSTED LANGEVIN ALGORITHMS

The Unadjusted Langevin Algorithm (ULA) (Robert et al., 1999) generates approximate samples from a target distribution with density $q(x) \propto \exp(-U(x))$ by discretizing the Langevin stochastic differential equation (SDE). Specifically, given a step-size $\eta_k > 0$, the ULA update is

$$x_{k+1} = x_k - \eta \nabla U(x_k) + \sqrt{2\eta_k}, \xi_k, \tag{15}$$

where $\xi_k \sim \mathcal{N}(0, I)$ are independent Gaussian noise. Due to discretization errors, ULA introduces sampling bias.

The Metropolis Adjusted Langevin Algorithm (MALA) (Robert et al., 1999) improves upon ULA by incorporating a Metropolis-Hastings correction step to ensure exact sampling from the target distribution $q(x)$. Given a current state $x_k$, MALA proposes a candidate $x'$ via

$$x' = x_k - \eta \nabla U(x_k) + \sqrt{2\eta}, \xi_k, \tag{16}$$

and accepts it with probability: $\alpha(x_k, x') = \min\left\{1, \frac{\pi(x')q(x_k|x')}{q(x_k)q(x'|x_k)}\right\}$, where $q(\cdot|\cdot)$ denotes the transition density induced by the proposal step. If rejected, the chain remains at $x_k$. This correction guarantees that the stationary distribution matches exactly the target distribution $\pi(x)$.

---

**Algorithm 3:** Leapfrog integrator $\eta_{\epsilon,L}$

---

1: **Input**: initial state $(x_0, v_0)$;
$\quad v_0 = v_0 - \frac{\epsilon}{2}\nabla U(x_0)$;
$\quad$ **for** $m = 0$ **to** $L - 1$ **do**
$\quad\quad x_{m+1} = x_m + \epsilon v_m$;
$\quad\quad v_{m+1} = v_m - \epsilon\nabla U(x_{m+1})$;
$\quad v_L = v_L + \frac{\epsilon}{2}\nabla U(x_L)$
2: **Output**: $(x_L, v_L)$

---

### B.2.3  HAMILTONIAN MONTE CARLO

Hamiltonian Monte Carlo (HMC) (Neal et al., 2011) is an accept–reject MCMC method for unnormalized continuous densities on $\mathbb{R}^d$ where partial derivatives of the log density exist. By associating the target variable with the position of a particle in space and the density with its potential energy, the method introduces an auxiliary momentum variable and implements Hamiltonian dynamics to achieve extensive exploration while maintaining a high acceptance probability.

Specifically, the unnormalized target density $q$ is expressed in canonical form $q(x) \propto e^{-U(x)}$, where $U(x) \triangleq -\ln q(x)$ represents the potential energy. The momentum variable $v$ gives the kinetic energy $K(v) \triangleq \frac{\|v\|^2}{2}$. This forms the Hamiltonian $H(x, v) = U(x) + K(v)$ with the associated joint distribution $\pi(x, v) \propto e^{-\left(U(x)+K(v)\right)}$, where $x$ and $v$ are considered independent with marginals $q$ and the standard Gaussian distribution respectively. HMC generates samples from $\pi$ with MCMC, where each chain iteration starts by resampling the momentum, $v' \sim \mathcal{N}(0, I)$, while keeping the position unchanged, $x' = x$. Then, a Metropolis update step is implemented by generating proposals using Hamiltonian dynamics

$$\frac{dx}{dt} = v, \quad \frac{dv}{dt} = -\nabla_x U \tag{17}$$

to propagate $(x', v')$ along trajectories of constant energy to $(x^*, v^*)$ and accepting the new state with probability $\alpha = \frac{\pi(x^*, v^*)}{\pi(x', v')}$.

Integrating equation 17 with the leapfrog method (Algorithm 3) ensures $\alpha \approx 1$, as $H$ is approximately constant and the transformation is volume-preserving. Still, the integration may move $x$ to positions with very different marginal density $U(x)$. The resampling step prevents the marginal $U$ from being constrained by the initial value of $H$. Thus, the momentum variable is critical for efficient exploration of the space. Algorithm 4 implements HMC when $q = q'_0$, initializing the process by $q_0$.

HMC can be tuned by appropriately choosing the step size and the number of leapfrog steps (Neal et al., 2011). It is generally advised to choose the parameters such that the empirical acceptance rate is around the optimal value of 65%. One may also randomly select these parameters from fairly small intervals at each Markov chain iteration to ensure that both big steps and fine-tuning steps can be taken at various points in the chain.

### B.2.4  OPTIMIZATION-BASED SAMPLING

To solve equation 3 or equation 5, we can use any preferred optimization algorithm such as stochastic gradient descent (SGD) or Limited-memory Broyden–Fletcher–Goldfarb–Shanno (L-BFGS). Using the torchdiffeq package, the gradient of the objective can be computed via automatic differentiation. With access to the gradient, we iteratively refine the initial sample using a standard update rule. Starting from an initial $x_0^{(0)}$, the update takes the form

$$x_0^{(k+1)} = \texttt{OPT\_Alg}(x_0^{(k)}),$$

where $\texttt{OPT\_Alg}$ denotes the chosen optimization routine. We can feed the final $x_0^K$ into $T^*$ to generate the sample $x_1 = T^*(x_0^K)$.

**Algorithm 4:** Hamiltonian Monte Carlo

1: **Input**: samples from $x_0 \sim q_0$;
    **for** $n = 0$ **to** $N - 1$ **do**
      $v'_n \sim N(0, I)$;
      $(x^*, v^*) = \eta_{\epsilon, L}(x_n, v'_n)$;
      $\alpha = e^{-\left(U(x^*) + K(v^*)\right) + U(x_n) + K(v'_n)}$;
      Draw $u \sim \mathcal{U}(0, 1)$;
      **if** $u < \alpha$ **then**
        $x_{n+1} = x^*$
      **else**
        $x_{n+1} = x_n$
2: Set $x'_0 = x_N$
3: **Output**: sample $x'_0$ from $q'_0$

### B.2.5 REGULARIZERS FOR NON-GAUSSIAN SOURCE DISTRIBUTIONS IN OPTIMIZATION-BASED SAMPLING

We first note for clarity that our theory (Theorems 1–2) holds for non-Gaussian source distributions $q_0$. While the regularization expressions we exemplify in the main paper are specific to the standard Gaussian distribution, the design procedure naturally extends to other choices of source distribution. Specifically, we formulate the optimization problem as $\min_x J(T(x)) + R(x)$, where the regularization term $R(x)$ should be small when $x$ lies in a high density region of $q_0$. Table 6 summarizes possible designs of $R(x)$ corresponding to non-Gaussian source distributions.

Table 6: Example designs of $R(x)$ beyond Gaussian source distributions

| Source distribution | Regularizer $R(x)$ | Intuition |
|---|---|---|
| Gaussian | $(\|x\| - r)^2$ | Keeps samples around a shell with radius $r$ |
| Anisotropic Gaussian $\mathcal{N}(0, \Sigma)$ | $\left(\sqrt{x^\top \Sigma^{-1} x} - r\right)^2$ | Keeps samples around a high-density ellipsoid. |
| Mixture of Gaussians | $\min_k [(x - \mu_k)^\top \Sigma_k^{-1} (x - \mu_k) - Tr(\Sigma_k)]^2$ | Union of component shells |
| Laplace | $\beta(\|x\|_1 - s)^2$ | Keeps samples around a typical l1-norm magnitude $s$ |
| Uniform (bounded support) | $\mathrm{dist}(x, B)^2$ | Keeps samples close to the support |
| Manifold prior | $\mathrm{dist}(x, M)^2$ | Keeps samples in or close to the manifold. |

## C ADDITIONAL EXPERIMENTAL DETAILS

### C.1 2D EXPERIMENTS

In this section, we present more details on the 2D example in Section 5.1, including implementation details and additional experimental results. All experiments were run on a single NVIDIA A100 GPU.

#### C.1.1 IMPLEMENTATION DETAILS

**Flow matching model:** The vector field is approximated using a time-varying multilayer perceptron (MLP) adopted from Tong et al. (2023). We train a standard vector field model using an independent coupling distribution $\pi = q_0 \times q_1$, and an optimal vector field model using the optimal

joint distribution $\pi^*$ in equation 1. Each model is trained for 20,000 epochs with a batch size of 256, employing the Adam optimizer.

**Conditional sampling:** We consider three pairs of source and target distributions: (i) 8-Gaussian to Moon, (ii) Uniform to 8-Gaussian, and (iii) Circle to S-Curve. For these three tasks, we respectively select loss functions $J(x) = ((x[2])^2)/0.4$, $J(x) = 4|x[1] + x[2]|$, and $J(x) = 5|x[1] - x[2]|$, where $x := (x[1], x[2])$.

**Implementation of D-Flow in Ben-Hamu et al. (2024):** Among several choices of regularization terms in D-Flow, we employ $-\ln q_1$, which ensures the generated samples stay close to the target distribution $q_1$. Although $q_1$ generally lacks an explicit form, in this 2D experiment, we approximate it using kernel density estimation. For the pre-trained vector fields used in D-Flow, we evaluate two variants: a standard model trained with an independent $\pi = q_0 \times q_1$ and an optimal vector field model trained with $\pi^*$ in equation 1. We refer to these variants as D-Flow and D-Flow-OT, respectively. In the optimization process, we use 60 optimization steps and employ SGD as the optimizer.

**Implementation of methods in Feng et al. (2025):** Among several training-free methods in Feng et al. (2025), we select two of the best methods g-sim-MC and g-MC, which perform well in low-dimensional settings. We use 100 and 50 Monte Carlo samples for g-sim-MC and g-MC, respectively. For both methods, the pre-trained model is selected as the optimal vector field.

**Implementation of SGFM:** We evaluate SGFM with four sampling methods: IS, ULA, MALA, and HMC. For ULA, MALA, and HMC, we run 100 sampling iterations. In both MALA and HMC, the proposal step-size is tuned to target an acceptance rate of 60%. Besides, each HMC iteration employs $L = 5$ leapfrog steps.

**Evaluation metric:** The sample quality is measured using the empirical 1-Wasserstein distance between the generated and ground truth distributions. In Figure 4, the generated distribution is estimated using 2000 samples, while the ground truth distribution is estimated using $10,000$ samples. In Figure 5, the generated distribution is instead estimated using an increasing number of samples as indicated by the $x$-axis. All experiments were conducted ten times, with solid lines and shaded areas representing the mean and standard deviation.

### C.1.2 IMPACT OF PRE-TRAINED VECTOR FIELDS ON GUIDANCE PERFORMANCE

In Section 5.1, we evaluate the impact of pre-trained vector fields on guidance performance. We provide more details here. The Lipschitz constant $L_v$ for each of the fields is empirically approximated by evaluating the derivative at 50000 uniformly selected points in the domain $[0, 1] \times \mathcal{X}$ using autodifferentiation. Here, $\mathcal{X}$ is the domain of the source and target samples. The guidance performance is evaluated using the Wasserstein distance between the true target distribution and the generated distribution. We use 10,000 samples from the target distribution as ground truth and generate 2,000 guided samples for evaluation. The guidance error (Wasserstein distance) is reported as mean $\pm$ standard deviation over 20 runs.

### C.1.3 ADDITIONAL RESULTS

We conduct an extensive comparison across different source and target distributions. The generated distributions are visualized in Figure 7. In the first experiment (8-Gaussian to moon), we observe that all the baseline methods and most Langevin-based algorithms struggle. The key reason is the highly multimodal landscape of this task, which makes the sampler easy to get trapped in local minima. However, SGFM-IS successfully navigates the posterior distribution. In the second experiment (circular to S-curve), many guidance methods perform well, but D-Flow tends to overemphasize minimizing the loss $J$ and loses sample diversity. In the third experiment (uniform to S-curve), we observe that D-Flow fails to generate a satisfactory conditional distribution, and g-MC exhibits slight deterioration in sample quality. Across every experiment, SGFM-IS consistently delivers high-quality, diverse samples, with SGFM-MALA and SGFM-HMC providing strong alternatives in the latter two tasks.

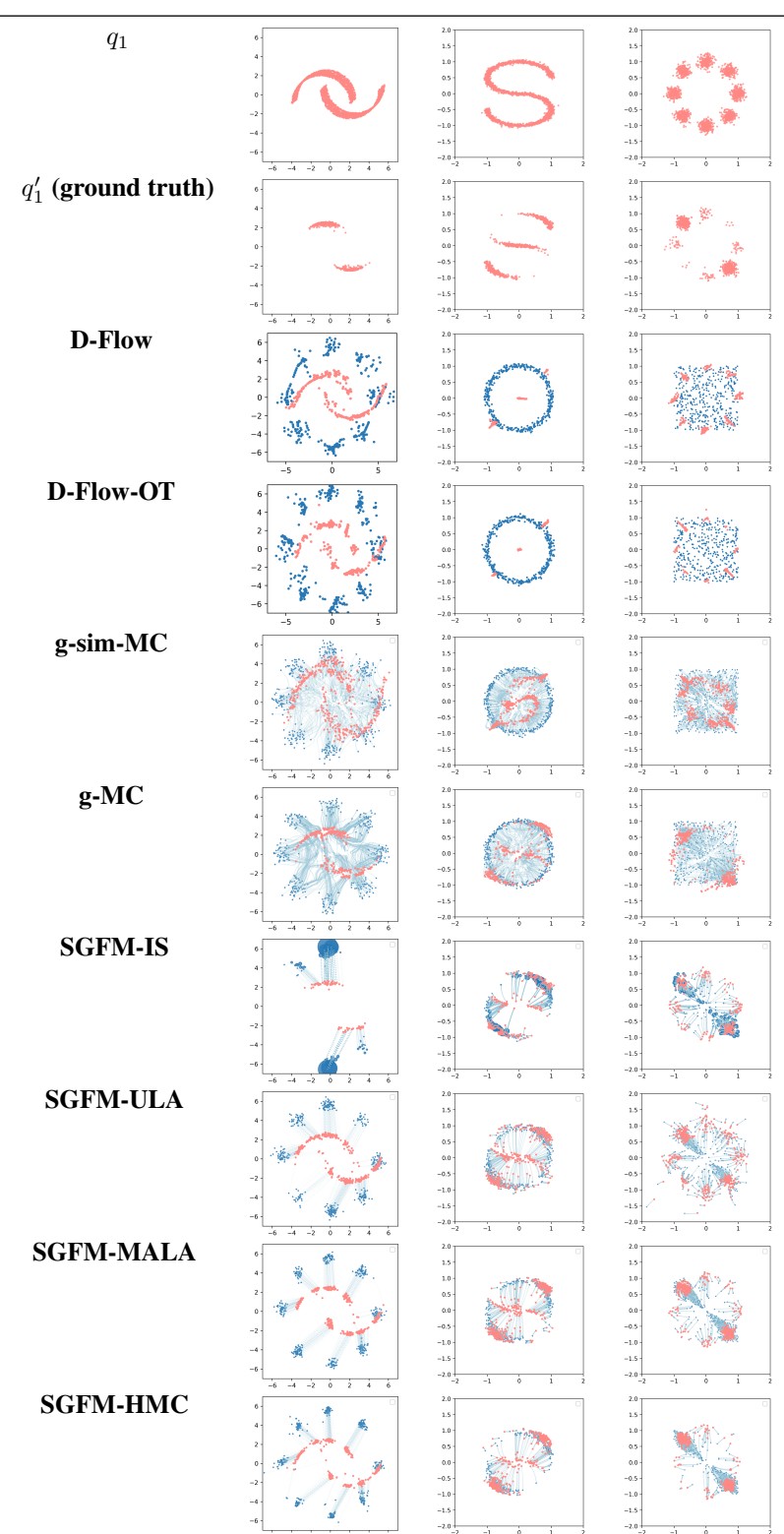

Figure 7: Results of synthetic datasets with different source distributions (marked in blue) and target distributions (marked in red).

We provide additional experiments that evaluate the sample quality with other pairs of source and target distributions. Since the distributions are simple and low-dimensional, we adopt IS as the sampling method and refer to our guidance method as SGFM-IS. Figure 8 presents the generation results with an 8-Gaussian source distribution and a moon target distribution. We observe that SGFM-IS achieves superior sample quality across varying running times. However, both baseline methods perform poorly and more running time did not help. The primary reason for failure is the multi-modal structure of this generation task, which makes samplers trapped in local optima, as illustrated in Figure 7. These results underscore the flexibility of our guidance framework, which allows for the tailored selection of advanced sampling strategies to suit different tasks.

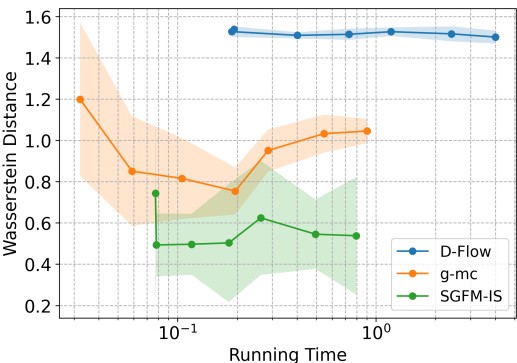

Figure 8: Comparison of sample quality and running time in 2D example, with an 8-gaussian source distribution and a moon target distribution.

Figure 9 presents the generation results with a circle source distribution and an S-curve target distribution. We observe that D-Flow performs even worse with increasing running time. As shown in Figure 7, D-Flow tends to overly transport points to the line $x[1] = x[2]$, indicating that D-Flow overemphasizes minimizing the loss $J$ and loses sample diversity.

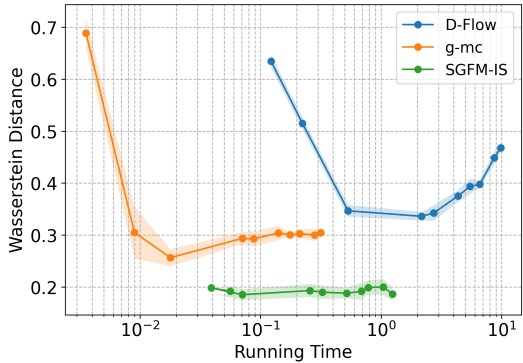

Figure 9: Comparison of sample quality and running time in 2D example, with a circle source distribution and an S-curve target distribution.

## C.2 PDE INVERSE PROBLEM

In this section, we provide more details on the physics-informed inverse problem in Section 5.2, including an outline of the Darcy flow equations, implementation details, and additional sample outcomes.

### C.2.1 DARCY FLOW EQUATIONS

Darcy flow is an elliptic PDE describing fluid flow through a porous medium,

$$
\begin{aligned}
\mathbf{u}(\mathbf{x}) &= -K(\mathbf{x})\nabla p(\mathbf{x}), & x \in \Omega \\
\nabla \mathbf{u}(\mathbf{x}) &= f(\mathbf{x}), & x \in \Omega \\
\mathbf{u}(\mathbf{x}) \cdot \hat{\mathbf{n}}(x) &= 0, & x \in \partial\Omega \\
\int_\Omega p(\mathbf{x})d\mathbf{x} &= 0,
\end{aligned}
\tag{18}
$$

where $K$ is the permeability field, $f$ is a source function, and $p$ is the resulting pressure field. In alignment with Bastek et al. (2024); Jacobsen et al. (2025), we consider the equations on a square domain $\Omega = [0,1]^2$ with resolution $64 \times 64$ and let $f$ be a constant function. In this setting, a dataset of pairwise solutions $(K, p)$ is offered by Bastek et al. (2024), which is generated by translating equation 18 to a linear system using finite difference approximations of the derivatives, and then solving this system.

### C.2.2 IMPLEMENTATION DETAILS

**Flow matching model:** The vector field defining the flow-matching model is approximated using a U-Net architecture adopted from Tong et al. (2023). The source distribution is a standard Gaussian distribution. In addition to the flow matching objective, the loss is regularized by the physics-residual following Bastek et al. (2024). The residual is computed using $\hat{x}_1$ from (Feng et al., 2025, Eq. 4) as data-space estimate and we select $\Sigma_t = \frac{1-t}{t}$ and $c = 10^{-2}$. The model is trained on $10^4$ samples for 200 epochs using the Adam optimizer with an initial learning rate $\eta = 10^{-4}$ which decays exponentially with a factor $\gamma = 0.99$.

**Conditional sampling:** All methods are initialized by the same set of samples from the unmodified source distribution. To balance the scale of the cost function $J$ and the prior probability $\log q_0$, the cost is scaled by a factor $\frac{1}{\lambda}$ where $\lambda = 10^{-3}$. To simulate a setting where true solutions are unavailable, all methods were tuned before observing the validity scores of the outcomes.

**Implementation of SGFM-HMC:** SGFM-HMC is implemented by running the HMC algorithm for $N_{HMC} = 100$ steps with $L = 3$ leapfrog steps, where the step size is randomly selected in each Markov chain iteration as $\epsilon = 5 \times \left(10^{-4} + \zeta \times 10^{-3}\right)$ where $\zeta \sim \chi^2(2)$ with $\chi^2(2)$ being the chi-squared distribution with two degrees of freedom. We found that this setting gives good acceptance ratios while allowing for a significant number of HMC iterations to be performed without having too long runtimes. The transport map is obtained by integrating the neural ODE associated with the vector field for two steps using the Dormand-Prince (Dopri5) method. We use the same transportation map both for the density computation in the HMC iterations and to map the sampled source point to the target space.

**Implementation of SGFM-OPT-1 and SGFM-OPT-2:** SGFM-OPT-1 (equation 4) and SGFM-OPT-2 (equation 5) are implemented using L-BFGS optimization with learning rate $\eta = 1$, maximum iterations of 20, and history size 100. The method is allowed to run for the same amount of runtime as HMC (which corresponds to approximately 15 optimization steps), but usually converges before that. The transport map is designed as in SGFM-HMC.

**Implementation of g-covA:** g-covA (Feng et al., 2025) is implemented using a linear schedule $\lambda_t^{\mathrm{covA}} = 10 \times \lambda$. We found that $\lambda_t^{\mathrm{covA}} = \lambda$ was not sufficient to observe a significant change in the guidance cost, while this choice achieves the lowest guidance cost of all methods. The ODE is integrated for three steps with the Dopri5 method, additional steps had no effect on performance.

### C.2.3 ADDITIONAL SAMPLES

To elaborate on the results in Figure 6, we present the pressure $\hat{p}$ sampled jointly with $\hat{K}$ for each of the outcomes in Figure 6 (where instead the true pressure $p_{\hat{K}}$ corresponding to $\hat{K}$ is shown) in Figure 10. Note that $\hat{p}$ serves as basis for the guidance signal since $p_{\hat{K}}$ is considered inaccessible. We observe that $\hat{p}$ generally aligns well with the target, while in Figure 6 many methods struggle with a large mismatch of $p_{\hat{K}}$ to target. This is because the process is driven to improbable pairs $(\hat{p}, \hat{K})$, leading to the mismatch between $p_{\hat{K}}$ and $\hat{p}$. This is also reflected by a poor validity score despite a low guidance cost. Finally, we present additional outcomes of $p_{\hat{K}}$ and $\hat{K}$ in Figure 11.

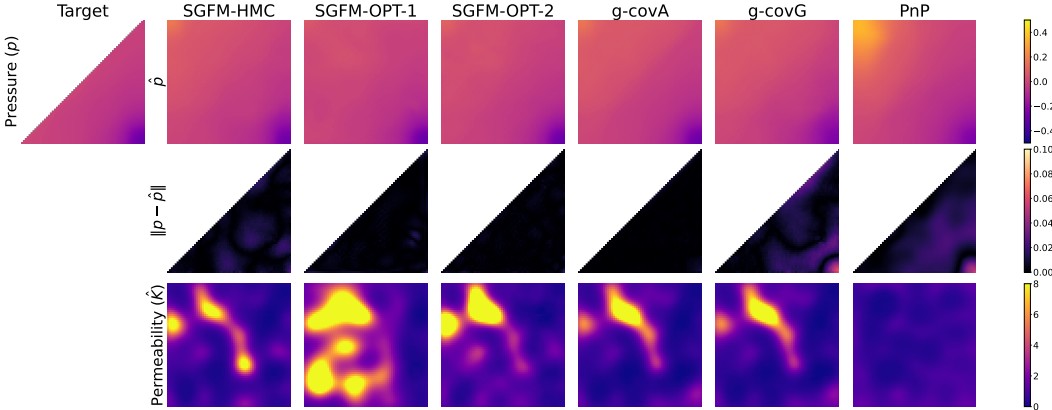

Figure 10: Solutions to the inverse problem of the Darcy flow equations. Top: target pressure and sampled pressure $\hat{p}$ (which is sampled jointly with inverse estimate $\hat{K}$ and serves as basis for the guidance signal); middle: target reconstruction error; bottom: inverse estimate $\hat{K}$ of the permeability field generated by conditional sampling.

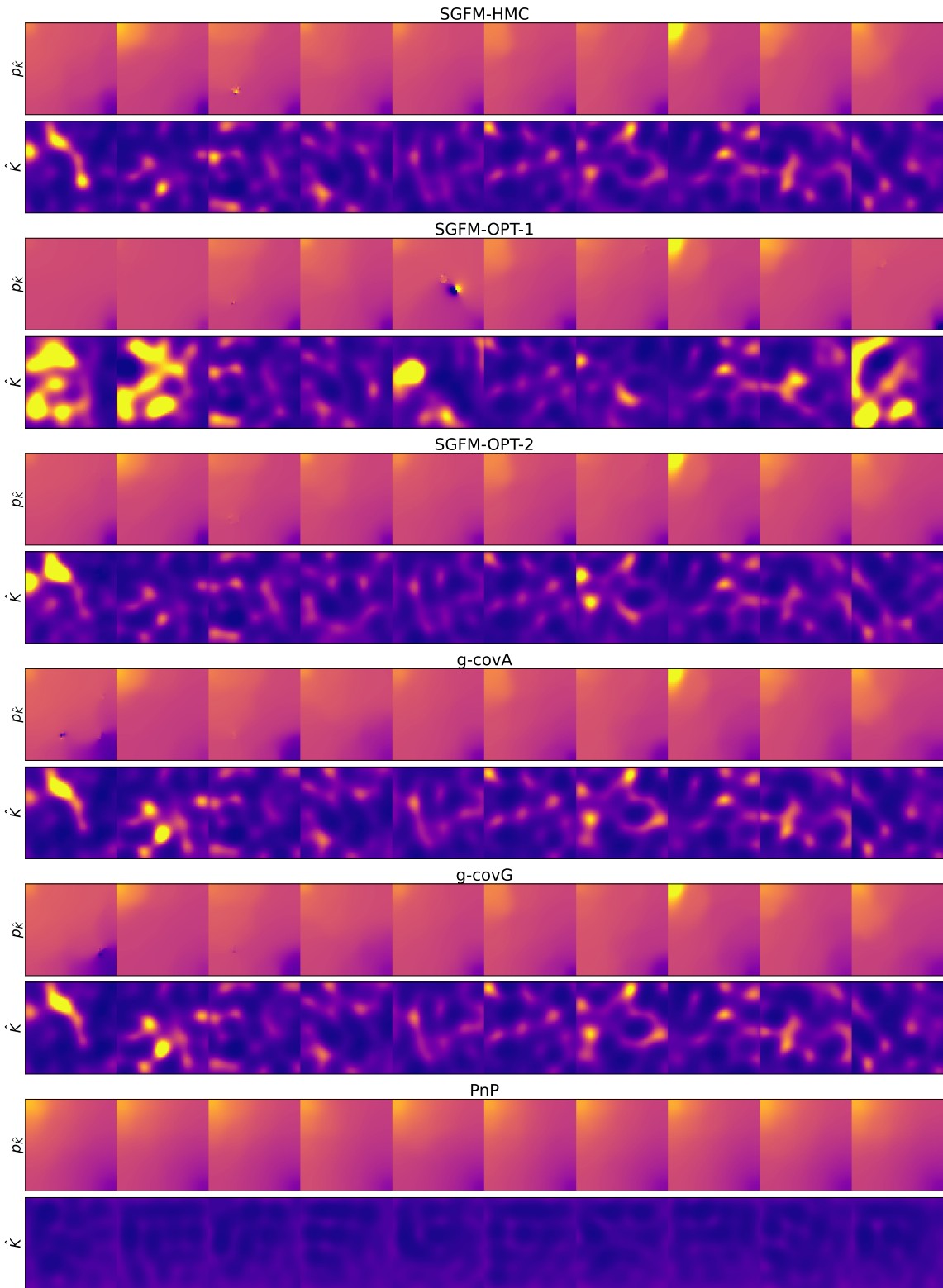

Figure 11: Solutions to the inverse problem of the Darcy flow equations - additional outcomes following the target in Figure 6. Top: true solution $p_{\hat{K}}$ corresponding to inverse estimate $\hat{K}$; bottom: inverse estimate $\hat{K}$ of the permeability field generated by conditional sampling.

### C.3 Image inverse problem on CelebA

#### C.3.1 Implementation details

The regularizers or constraints used in different variants of optimization-based sampling were summarized in Table 3. For regularization-based methods, we introduce a weighting coefficient and tune it to achieve optimal performance. For the constraint-based method, we project the solution onto the hyperspherical shell after each update.

### C.3.2   GENERATED CELEBA SAMPLES

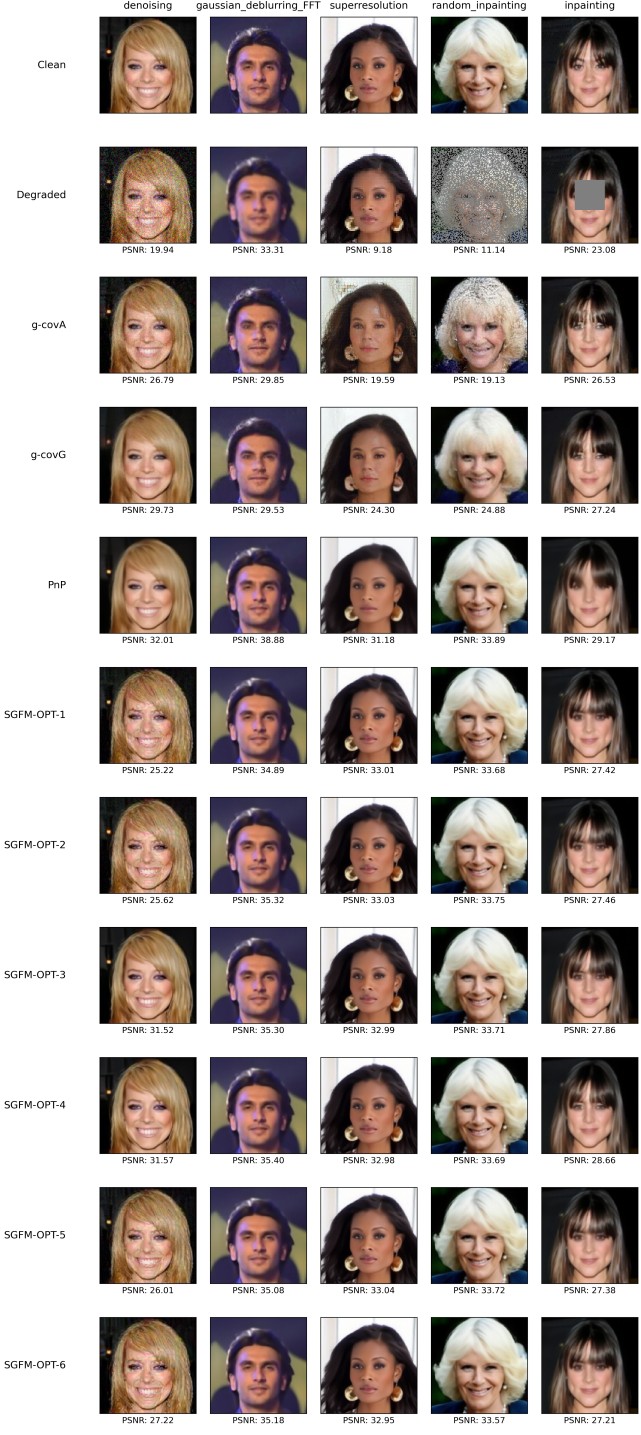

Figure 12: Comparison of image restoration methods on CelebA.

## C.4 MNIST IMAGE GENERATION

### C.4.1 CONDITIONAL GENERATION ON MNIST

We perform conditional image generation experiments on the MNIST dataset, where the generated samples are conditioned on provided labels. Given a target label, the loss function $J$ corresponds to the negative log-likelihood of the label computed via a classifier. We select ULA as our sampling method and benchmark its performance against the baseline method g-covA in Feng et al. (2025).

| Method | SGFM-ULA | g-covA |
|---|---|---|
| **Accuracy** | 87.6% | **98.5%** |
| **FID** | **46.7** | 57.1 |

Table 7: The label accuracy (higher is better) and FID (lower is better).

Performance is evaluated using the Fréchet Inception Distance (FID) and label accuracy. A separate classifier determines accuracy to avoid overconfidence. The experimental results, detailed in Table 7, indicate that while our proposed method yields relatively lower label accuracy, it achieves a superior FID score. Figure 13 presents illustrative examples of generated images from both methods. We observe that although g-covA consistently generates images corresponding to the correct digits, the generated samples exhibit limited diversity, characterized by uniformly thick strokes and similar visual styles. In contrast, the ground-truth MNIST distribution inherently comprises digits exhibiting diverse shapes, styles, and stroke widths. Our approach demonstrates improved sample diversity over g-covA (although with some degradation in quality compared to ground truth). This gives a better covariance match between the empirically generated distribution and the real data distribution, resulting in a lower FID score.

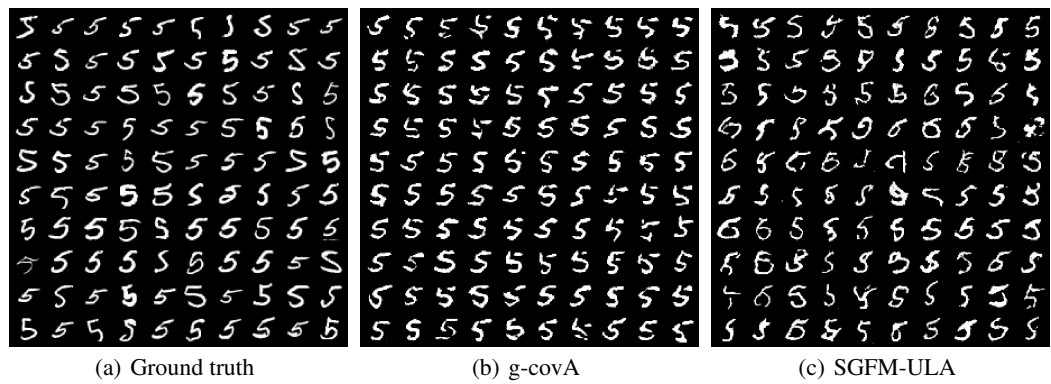

(a) Ground truth      (b) g-covA      (c) SGFM-ULA

Figure 13: MNIST sample generation conditioned on digit 5.

### C.4.2 IMPLEMENTATION DETAILS

**Pre-trained models:** For the classifier used for guidance, we train a convolutional neural network classifier on MNIST, which achieves an accuracy of 96.8% on the standard test set. For the classifier used for evaluating the accuracy of generated samples, we adopt an independent pre-trained Vision Transformer classifier[1], which achieves higher robustness with an accuracy of 98.7% on the testing distribution. Following Tong et al. (2023), the vector field model used in our experiments is trained using a U-Net architecture initialized from a Gaussian distribution. It was trained for three epochs, each consisting of 468 iterations.

**Conditional generation:** The objective of this task is to generate images conditioned on a specified label and stay close to the original dataset. The guidance for this conditional generation utilizes a loss function $J$ defined as the negative log-probability of the targeted label $i$:

$$J(x) = -\log \operatorname{softmax}(h(x))_i,$$

where $h(x)$ represents the logits returned by a pre-trained classifier. To balance the scale of the cost function and the probability density function, we scale the loss function $J$ by $\frac{1}{\lambda}$ with $\lambda = 10^{-3}$.

---

[1]https://github.com/sssingh/hand-written-digit-classification/tree/master

**Implementation of the method in Feng et al. (2025):** We select g-covA as the baseline method in Feng et al. (2025), which shows superior performance in image problems. We use a constant schedule $\lambda_t^{\text{cov-A}} = \lambda^{\text{cov-A}}$. The ODE is integrated for 100 steps using the Dopri5 method.

**Implementation of SGFM-ULA:** SGFM-ULA is implemented by running the ULA algorithm over a maximum of 150 steps with a batch size of 16. The step size in each iteration is selected as $5 \times 10^{-4} \times \zeta$, where $\zeta \sim \chi^2(2)$ with $\chi^2(2)$ being the chi-squared distribution with two degrees of freedom. Each batch takes about 218 seconds to process.

**Evaluation metric:** We assess the quality of generated images using the Fréchet Inception Distance (FID), which measures the similarity between two distributions based on their means and covariances. The reference images used in the FID calculation are subsets of the MNIST training dataset corresponding to each target label, and evaluations are performed using 400 samples.

## C.5 ODE FORWARD PROBLEM

In this section, we present another example of conditional generation in the context of physics-informed generative modeling. Instead of the Darcy flow equations (equation 18), we consider an ODE with truncated solution trajectories. Compared to the problem in Section 5.2, this example has lower dimension which allows us to further explore how well the SGFM framework and benchmark methods approximate the target distribution.

### C.5.1 CONDITIONAL GENERATION OF ODE TRAJECTORIES VIA FLOW MATCHING

Consider the ODE

$$\dot{x}(t) = -\theta_a x(t) + \theta_b \sin(\theta_\omega t), \quad x(0) = 0, \quad x(t) \in \mathbb{R}, \tag{19}$$

where a flow matching model is trained to sample from the joint distribution of the ODE parameters $[\theta_a, \theta_b, \theta_\omega] \triangleq \theta \sim \mathcal{U}(1,3)^3$ and the set of corresponding discretized solutions $x_\theta \in \mathbb{R}^{100}$. The source distribution is a standard Gaussian distribution for the parameters $\theta$ and a Gaussian Process with zero mean and squared exponential kernel for $x_\theta$ to encourage smooth solutions.

The conditional sampling problem is to generate solution trajectories consistent with a partial observation of the ODE parameters $[\theta_a^*, \theta_b^*, \cdot]$, which defines a family of admissible solutions $\{x_{\theta_a^*, \theta_b^*, \cdot}\}_{\theta_\omega \in [1,3]}$. The cost function is the reconstruction error of the target parameters, $J(\theta, x_\theta) = \|\theta_a - \theta_a^*\|^2 + \|\theta_b - \theta_b^*\|^2$. This corresponds to a soft constraint on $\theta_a$ and $\theta_b$, since the marginal target distributions for $\theta_a$ and $\theta_b$ then behave like posteriors with uniform priors and Gaussian likelihoods. Figure 14 shows samples from the unconditional model and conditional samples using SGFM-HMC, SGFM-OPT (Ben-Hamu et al., 2024) corresponding to equation 3[2], and g-covA (Feng et al., 2025).

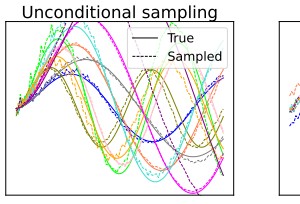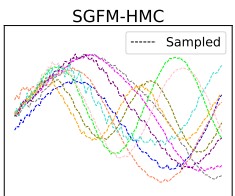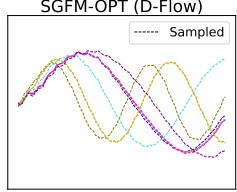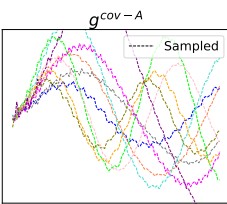

Figure 14: Solutions to the forward ODE problem. Samples $[\theta, x_\theta] \in \mathbb{R}^{103}$ consist of ODE parameters $\theta \sim \mathcal{U}(1,3)^3$ and corresponding solution trajectories $x_\theta \in \mathbb{R}^{100}$. The conditioning set consists of a partial observation of the ODE parameters, $[\theta_a^*, \theta_b^*, \cdot]$, which yields a family of admissible solutions $\{x_{\theta_a^*, \theta_b^*, \cdot}\}_{\theta_\omega \in [1,3]}$.

We evaluate the methods by generating $10^3$ samples and assessing both their physical consistency and how well the empirical distribution approximates the target distribution. The latter is evaluated by comparing the parameter outcomes to equal-tailed credible intervals derived from the target distribution, which are obtained by MCMC simulation. The results in Figure 15 show that SGFM-based methods generate samples of higher physical consistency compared to g-covA. Furthermore, SGFM-HMC achieves the most representative distribution over the parameters. In contrast, SGFM-OPT collapses to the modes for the conditioned parameters $\theta_a, \theta_b$, and importantly fails to capture the full admissible range of the unconditioned parameter $\theta_\omega$. g-covA, on the other hand, captures the full range of admissible $\theta_\omega$ but excessively generates values outside of the credible intervals, sometimes outside of the training data distribution. Thus, we conclude that SGFM-HMC best approximates the target distribution. Together with the PDE inverse problem in Section 5.2, this illustrates the trade-off between asymptotic exactness and computational complexity that the methods within SGFM can balance: SGFM-HMC yields a more accurate empirical distribution, while SGFM-OPT methods are faster at the expense of local mode collapse.

---

[2]In this example, the source distribution is not directly compatible with equation 4-equation 5. However, equation 3 remains applicable, which reflects the base implementation in D-Flow.

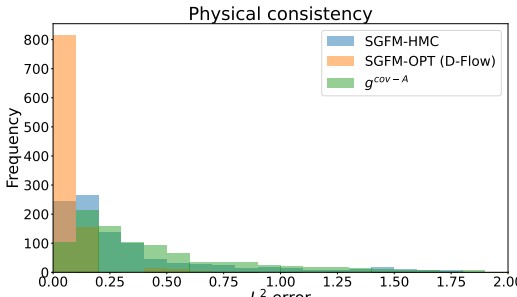 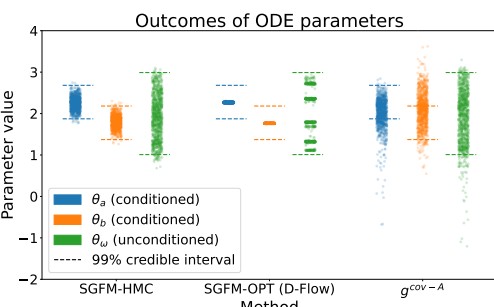

Figure 15: Physical consistency of samples and closeness of the empirical distribution to the target distribution. The physical consistency is measured by the relative $L^2$ error between the sampled trajectory and the true trajectory under the jointly sampled ODE parameters. Each method's ability to capture the target distribution is then assessed by analyzing the empirical distribution of the sampled ODE parameters. In the ideal case, 99% of the samples fall within the credible interval indicated by the dashed lines. The true marginal target distributions for the conditioned parameters $\theta_a, \theta_b$ behave like posteriors with a uniform prior and Gaussian likelihood, so the outcomes are expected to distribute smoothly across the bounds. Similarly, the true marginal target distribution for the unconditioned parameter $\theta_\omega$ is a uniform distribution, so the outcomes are expected to be distributed uniformly across the bound.

### C.5.2 IMPLEMENTATION DETAILS

**Flow matching model:** The vector field defining the flow-matching model is approximated using an MLP similar to Tong et al. (2023) with four hidden layers of size 256 and SELU activation functions. Furthermore, we add a Gaussian smoothing filter with non-learnable parameters on the last layer to encourage smooth solutions. The source distribution is a standard Gaussian distribution for the ODE parameters and a Gaussian process with zero mean and squared exponential kernel having length scale $l = 1$. The dataset consists of $10^4$ samples which are generated by sampling $\theta \sim \mathcal{U}(1,3)^3$ and integrating equation 19 for $t \in [0,5]$ using the Euler method with $\Delta t = 0.05$. The model is trained for $10^3$ epochs using the Adam optimizer with learning rate $\eta = 10^{-3}$.

**Conditional sampling:** All methods are initialized by the same set of samples from the unmodified source distribution. To balance the scale of the cost function $J$ and the prior probability $\log q_0$, the cost is scaled by a factor $\frac{1}{\lambda}$ where $\lambda = 5 \times 10^{-2}$.

**Implementation of SGFM-HMC:** SGFM-HMC is implemented by running the HMC algorithm for $N_{HMC} = 200$ steps with $L = 50$ leapfrog steps, where the step size is randomly selected in each Markov chain iteration as $\epsilon = 10^{-4}(1 + \zeta \times 15)$ where $\zeta \sim \chi^2(2)$ with $\chi^2(2)$ being the chi-squared distribution with two degrees of freedom. We found that this setting gives good acceptance ratios while allowing for a significant number of HMC iterations to be performed without having too long runtimes. The transport map is obtained by integrating the neural ODE associated with the vector field for two steps using the Dormand-Prince (Dopri5) method. We use the same transportation map both for the density computation in the HMC iterations and to map the sampled source point to the target space.

**Implementation of SGFM-OPT:** SGFM-OPT is implemented using L-BFGS optimization with learning rate $\eta = 1$, maximum iterations of 20, and history size 100. The method is allowed to run for approximately the same amount of runtime as HMC (which corresponds to 160 optimization steps), but usually converges before that. The transport map is designed as in SGFM-HMC. We use the same transportation map both for the density computation in the D-Flow iterations and to map the sampled source point to the target space.

**Implementation of g-covA:** g-covA (Feng et al., 2025) is implemented using a constant schedule $\lambda_t^{\text{covA}} = \lambda$. This choice cancels out the loss scaling factor $\frac{1}{\lambda}$, and other alternatives either degrade

physical consistency or impose weak constraints on the conditioned parameters. The ODE is integrated for three steps using the Dopri5 method, additional steps had no effect on performance.

## D    LIMITATIONS AND FUTURE WORK

Similar to other guidance methods that optimize the source samples, one limitation of our method lies in the long runtime due to the need to backpropagate through the ODE. Consequently, it would be interesting to incorporate efficient backpropagation in our framework. Additionally, training the optimal vector field requires access to the OT coupling, which becomes particularly challenging in high-dimensional settings. Although we can approximate $\pi^*$ using mini-batch data or entropic OT solvers, these approximations can introduce bias and may not scale well. Developing a more efficient and scalable approach to training the optimal vector field is another important avenue for future research.

This paper employs Large Language Models (LLMs) exclusively for the purpose of polishing written text.

