# OpenReview forum: "Source-Guided Flow Matching"
_ICLR.cc/2026/Conference — ICLR 2026 Poster_

### Official Review · Reviewer_wX5E · 2025-10-29

**Soundness:** 2
**Presentation:** 3
**Contribution:** 2
**Rating:** 4
**Confidence:** 3

**Summary:**

The paper proposes an interesting method to do guidance in conditional generation of flow matching, by modifying the source distribution instead of training the velocity neural network. The method has rigorous theoretical guarantee and various sampling algorithms can be utilized to sample from the modified source distribution, but this process can be very computationally expensive. Experimental comparisons with previous methods demonstrate the effectiveness of the proposed method.

**Strengths:**

The paper is well organized and easy to follow, motivated by an interesting theorem that to modify the source distribution can also achieve guidance. Building on this, the authors discuss several sampling algorithms to sample from the modified source distribution. Extensive experiments are also conducted to verify the the efficiency of the proposed framework.

**Weaknesses:**

1. The major concern is how to sample from $q_0'\propto q_0\cdot\exp(-J\circ T^* )$ efficiently?
The authors mention HMC and optimization-based methods, but all of these methods require gradient computation of the potential function $J\circ T^* $, where  $ T^* $  is the FM model. This requires backpropagatio through an ODE trajectory, which is super computationally expensive and impractical.

2. Following the question above, what if the potential function $J$ is not differentiable or to compute the gradient of $J$ is very expensive? In this case, only importance-sampling based algorithms can be used and the convergence is very slow. Will the proposed method be more efficient than fine-tuning velocity neural network using, say, reinforcement learning methods?

**Questions:**

Please see weakness part.

---

> ### Author Response · Authors · 2025-11-19
>
> Thank you for the helpful review and for noting that the paper is well-organized. We address the raised weaknesses below.
>
> > Efficient sampling from $q_0'$.
>
> We note that it is common for training-free guidance methods to require backpropagation through a partial or full trajectory (see e.g., [R1,R2,R3,R4]). One of our key insights is that the efficiency of our guidance methodology is greatly improved using mini-batch optimal transport (OT) training [R6], since the straight vector fields require fewer discretization steps to integrate the trajectory $t\in [0,1]$. Theorem 2 reflects this property as the log-Wasserstein error is scaled by the Lipschitz constant of the vector field, $L_v$, and it is experimentally verified in Appendix C.1.2. For example, optimal performance in the high-dimensional CelebA experiment is achieved using just 3 integration steps (Table 5 in the paper). In terms of inference time, SGFM-OPT variants processes a CelebA image in 1.5 minutes, markedly outperforming D-Flow, which requires 12.5 minutes for super-resolution and 15.5 minutes for deblurring (both tested on the same V100 GPU). Furthermore, the low-dimensional experiments in Figure 4,8,9 show that our methods have significantly higher efficiency than baseline methods. Finally, to further mitigate efficiency issues, our framework is compatible with advanced techniques such as the adjoint method seen in [R4] which reduces the memory complexity of backpropagation.
>
> > What if J is non-differentiable? Will the proposed method be more efficient than fine-tuning velocity neural network using RL?
>
> Efficient guidance for non-differentiable rewards, to the best of our knowledge, remains a significant challenge in flow matching and diffusion models. While experiments in this paper use differentiable rewards, SGFM readily accepts non-differentiable rewards and discrete distributions, and the proof is derived with minor modifications (Appendix A.1). Practically, SGFM is flexible to incorporate advanced sampling techniques for non-differentiable rewards, such as elliptical slice sampling [R5]. Regarding reinforcement learning-based fine-tuning approaches, we emphasize that these are training-based, while SGFM is training-free, which makes them not directly comparable.
>
>
>
>
> [R1] Feng R, et al. On the guidance of flow matching. arxiv. 2025.
>
> [R2] Song J, et al. Loss-guided diffusion models for plug-and-play controllable generation. arxiv. 2023.
>
> [R3] Ben-Hamu H, et al. D-flow: Differentiating through flows for controlled generation. arxiv. 2024.
>
> [R4] Wang L,et al. Training free guided flow matching with optimal control. arxiv. 2024.
>
> [R5] A Kalaivanan, et al. ESS-Flow: Training-free guidance of flow-based models as inference in source space. arxiv. 2025
>
> [R6] Tong A, et al. Improving and generalizing flow-based generative models with minibatch optimal transport. arxiv. 2023.

---

### Official Review · Reviewer_Tm1L · 2025-10-31

**Soundness:** 3
**Presentation:** 2
**Contribution:** 3
**Rating:** 6
**Confidence:** 4

**Summary:**

This paper presents the Source-Guided Flow Matching (SGFM) framework that modifies the source distribution directly to incorporate the guidance for flow matching. This framework is backed by the theorem that sampling from the modified source distribution and driving the flow along the exact vector field precisely recovers the desired target distribution. Based on this, the paper also provides the Wasserstein convergence guarantee. The framework is open for different sampling methods to sample from the modified source distribution. Finally, the paper experiments on synthetic 2D benchmarks, physics-informed generative tasks and imaging inverse problems to demonstrate the effectiveness and flexibility of the proposed method.

**Strengths:**

•	The paper is well-written and clearly-organized. The center of the proposed method based on modifying the source distribution is focused with in-depth discussion.

•	The framework is flexible to combine with different types of sampling methods based on the characteristics of the target distribution. The pros and cons of different methods are discussed in Section 4.

•	The experiments are selected from low to high dimensional problems, which is comprehensive.

**Weaknesses:**

•	The clarity of the notations could be improved. For example, it would be better to denote samples from $q’_0$ by $x’_0$ rather than $x_0$ in Algorithm 1 to reduce ambiguity.

•	Even though the framework is flexible to incorporate different types of sampling methods, the mode collapse issue when the target distribution is highly non-concave still remains. In addition, the remedy to mitigate the mode collapse issue in line 309 only adapts to the Gaussian source distribution, it remains unknown whether it could extend to general source distributions.

•	It would be beneficial to add the variance interval in Table 2 similar to Table 1.

**Questions:**

•	Is it possible to address the mode collapse issue with annealed sampling or add some other types of regularization term to equation (4) similar to (5)? What kind of regularization term should we use when the source distribution is not Gaussian?

•	The proposed method requires additional sampling procedure from the modified target distribution compare to the standard flow matching method. What is the benefit of the proposed method compared to the guided flow matching methods like [1] that does not require the additional sampling procedure?

•	[1] Xie, T., Zhu, Y., Yu, L., Yang, T., Cheng, Z., Zhang, S., ... & Zhang, C. (2024). Reflected flow matching. arXiv preprint arXiv:2405.16577.

---

> ### Author Response · Authors · 2025-11-19
>
> Thank you for your positive assessment and insightful feedback. We address the weakness and questions raised by you below.
>
> > Notation
>
> Thank you for highlighting ambiguity in our notation. We will improve the clarity following your advice.
>
> > Mode collapse as a limitation of SGFM
>
> Mode collapse is not an inherent limitation of SGFM. For example, we show that SGFM-IS (Fig 7) and SGFM-HMC (Fig 15) captures the full range of the target distribution. Generally, SGFM allows the user to balance exactness and computational efficiency according to their priorities through the choice of source distribution sampler. If the goal is to generate an empirical representation of the target distribution, we recommend exact samplers like IS and HMC. However, if the goal is to generate a few samples at low computational cost, we recommend the approximate optimization-based sampling methods. Crucially, SGFM supports both alternatives side-by-side, demonstrating its practical flexibility.
>
> > Mode collapse remedies beyond Gaussian distributions
>
> We first note for clarity that our theory (Theorems 1-2) holds for non-Gaussian source distributions $q_0$. While the regularization expressions we exemplify in the manuscript indeed are specific to the standard Gaussian distribution, the design procedure naturally extends to other choices of source distribution. Specifically, we formulate the optimization problem as $\min_x J(T(x)) + R(x)$, where the regularization term $R(x)$ should be small when $x$ lies in a high density region of $q_0$. The table below summarizes possible designs of $R(x)$ corresponding to non-Gaussian source distributions.
>
> Table 1: Example designs of $R(x)$
>
> | Source dist | Regularizer $R(x)$ | Intuition |
> | --- | --- | --- |
> | Gaussian | $(\|x\| - r)^2$ | Keeps samples around a shell with radius $r$. |
> | Anisotropic Gaussian $\mathcal{N}(0,\Sigma)$ | $\big(\sqrt{x^\top \Sigma^{-1} x} - r\big)^2$ | Keeps samples around a high-density ellipsoid. |
> | Mixture of Gaussians | $\min_k \big[(x - \mu_k)^\top \Sigma_k^{-1} (x - \mu_k) - \mathrm{Tr}(\Sigma_k)\big]^2$ | Union of component shells. |
> | Laplace | $\beta(\|x\|_1 - s)^2$ | Keeps samples around a typical $\ell_1$-norm magnitude $s$. |
> | Uniform (bounded support) | $\mathrm{dist}(x,B)^2$ | Keeps samples close to the support. |
> | Manifold prior | $\mathrm{dist}(x,M)^2$ | Keeps samples in or close to the manifold. |
>
> > Variance interval in Table 2
>
> We thank the reviewer for pointing out the need for variances in Table 2 (CelebA experiment). The experiment was repeated for 10 different images, and the mean and variance is reported in Table 2 below. For reference, we recall the OPT variants (Table 4 in the paper). The results show that the mean values align with the previous results and the variances are small. We will update the final version of the paper with this information.
>
> Table 2: PSNR comparison (mean±std)
>
> | Method | denoise   | deblur  | super-res    | random-inpaint | box-inpaint |
> |--------|-------|-------|-------|-------|-------|
> | g-covA | 26.7±0.4 | 28.5±1.8 | 19.9±1.1 | 20.8±1.5 | 26.0±2.7 |
> | g-covG | 30.7±1.4 | 28.3±1.9 | 24.6±1.5 | 25.0±1.6 | 28.3±1.9 |
> | PnP    | 32.6±1.5 | 36.9±3.2 | 32.0±1.9 | 34.1±2.9 | 31.1±2.9 |
> | SG1    | 30.3±2.3 | 34.0±2.3 | 34.0±2.6 | 33.6±2.7 | 31.3±2.5 |
> | SG2    | 30.4±2.2 | 34.1±2.3 | 34.1±2.6 | 33.6±2.8 | 31.2±2.6 |
> | SG3    | 32.2±1.2 | 34.1±2.3 | 34.1±2.6 | 33.6±2.8 | 30.7±2.4 |
> | SG4    | 32.3±1.3 | 34.2±2.4 | 34.1±2.6 | 33.6±2.8 | 30.4±2.2 |
> | SG5    | 30.3±2.2 | 34.1±2.4 | 34.1±2.6 | 33.6±2.8 | 31.2±2.6 |
> | SG6    | 31.2±1.5 | 33.9±2.2 | 33.8±2.4 | 33.4±2.6 | 31.1±2.6 |
>
>
> > Benefit compared to [1]
>
> In [1], the probability flow is guided to remain in a predefined domain $\Omega$ while integrating over $t\in[0,1]$. Guidance is achieved by reflecting the velocity field at the boundary $\partial\Omega$. For example, in the image generation experiments, $\Omega$ is a bounded hypercube, which ensures image fidelity (e.g., by preventing oversaturation) in the final generated images. In SGFM, we consider guided sampling in the sense that we exactly target a tilted distribution in closed form, $q_1'(x_1')\propto q_1(x_1')e^{-J(x_1')}$. For complex target distributions such as in our experiments, it is not obvious how to design $\Omega$ such that the flow exactly terminates in $q_1'$. Specifically, for every $t\in[0,1]$, we need to specify when $x_t$ hits $\partial \Omega$ and how it should be reflected. Thus, our approach is designed for a different type of guidance, such that the flow is exactly steered to the closed-form target distribution rather than constrained to a safe region. Finally, while [1] includes an experiment on class-conditional generation of images (akin to our experiments), the experiment achieves class-conditioning using a pre-trained conditional score function (with $\Omega$ a bounded hypercube to ensure image fidelity); in contrast, we offer a training-free approach to conditional sampling.

---

### Official Review · Reviewer_T7yj · 2025-11-04

**Soundness:** 3
**Presentation:** 3
**Contribution:** 3
**Rating:** 6
**Confidence:** 2

**Summary:**

The paper presents an approach for guided sampling using flow matching by changing the source distribution while keeping the pretrained vector field intact. The authors theoretically derive the form of such a source distribution. Building on this, the authors derive the bound for the Wasserstein error in generative modeling and show that it is linear in terms of the error in sampling from the source distribution. Based on this finding, the authors propose various strategies for accurately sampling from the source distribution. Through empirical analysis on 2d synthetic, physics-based Darcy flow, and natural image datasets, they show the effectiveness of their approach compared to the baselines.

**Strengths:**

I appreciate the thorough theoretical analysis in presenting the method. The writing is clear, and all the theory was backed by intuition. The idea of modifying the source distribution through the constraint function is interesting and has not been explored as far as I am aware.

**Weaknesses:**

I am not very familiar with the baselines that have been compared with. The main weakness of this method is the weak empirical performance on the Darcy flow data. The promise of this method, as I see it, is in enforcing constraints (differentiable) in the generative modeling. Therefore, physical consistency is an important metric in which the method falls short compared to PnP. This is not a major weakness; however, I would request the authors to add a discussion on this.

**Questions:**

1. In line 91, please define the composition operator $\circ$.

2. For Darcy flow data, how was the conditional sampling done based on a partial pressure field? Did the authors use masking in the ODE reverse process to achieve this?

3. Please explain the usefulness of the "validity of the inverse estimate" metric when it is observed that the joint sampling matches closely with the target pressure field.

---

> ### Author Response · Authors · 2025-11-19
>
> Thank you for your thoughtful and constructive review. We address the weakness and questions that you raised below.
>
> > Performance of SGFM versus PnP on Darcy flow
>
> We recall that the prior model jointly generates permeability $\hat{K}$ and pressure $\hat{p}$ fields as solutions of the Darcy flow PDE, and we consider constrained sampling in the sense that $\hat{K}$ aligns with the partial pressure observations $y$ as an inverse estimate. This requires that 1) the sampled pressure $\hat{p}$ matches the observations $y$ at the corresponding spatial locations, and 2) $\hat{K}$ remains physically consistent with $\hat{p}$. While PnP achieves a good physical consistency, it fails to align $\hat{p}$ with the observations $y$ as reflected by the poor guidance cost (Table 1, recalled below). In contrast, SGFM-HMC and SGFM-OPT-2 generate samples that match the observations more closely, while preserving the level of physical consistency expected from the prior model ("Unconditional sampling" in Table 1). Thus, samples from these SGFM methods yield better inverse estimates $\hat{K}$ compared to those generated by PnP (as reflected by the validity scores).
>
> Table 1: Performance of guidance methods in the Darcy flow inverse problem
>
> | **Method**                 | **Validity of Inverse Estimate $(\downarrow)$** | **Guidance Cost $(\downarrow)$**      | **Physical Consistency $(\downarrow)$** |
> |----------------------------|---------------------------------------|-----------------------------|-------------------------------|
> | SGFM-HMC                   | 0.591 [0.532, 0.654]                  | 0.281 [0.248, 0.335]        | 0.188 [0.168, 0.228]          |
> | SGFM-OPT-1                 | 0.907 [0.503, 1.875]                  | 0.206 [0.149, 0.294]        | 0.421 [0.174, 0.770]          |
> | SGFM-OPT-2                 | 0.474 [0.416, 0.562]                  | 0.187 [0.131, 0.218]        | 0.194 [0.157, 0.215]          |
> | g-covA                     | 0.992 [0.857, 1.293]                  | 0.030 [0.028, 0.052]        | 0.289 [0.247, 0.351]          |
> | g-covG                     | 0.955 [0.814, 1.201]                  | 0.242 [0.163, 0.388]        | 0.245 [0.190, 0.285]          |
> | PnP                        | 1.055 [0.950, 1.204]                  | 0.610 [0.590, 0.632]        | 0.116 [0.099, 0.129]          |
> | Unconditional sampling     | 1.006 [0.860, 1.269]                  | 1.051 [0.905, 1.289]        | 0.214 [0.167, 0.274]          |
>
> > Missing definition of composition operator
>
> Thank you for pointing out the missing definition of the composition operator, $f\circ g(x) :=f(g(x))$. We will ensure to add it in the final version.
>
> > How do we condition on the partial pressure field?
>
> We condition on the partial pressure observations $y$ by defining the corresponding task-specific loss $J$ and applying SGFM to tilt the generative model. Specifically, with $x_1=[\hat{K}, \hat{p}]$ and $q_1$ the prior model, the conditional target distribution is defined as $q_1'(x_1)\propto q_1(x_1)e^{-J(x_1)}$ where $J(x_1)=\|y - H\odot\hat{p}\|$ measures the mismatch between $y$ and the sampled pressure $\hat{p}$ evaluated at the corresponding grid points $H\in\{0,1\}^{64\times 64}$ (with $\odot$ denoting the Hadamard product). To sample from $q_1'$, we follow the SGFM methodology by sampling from the corresponding modified source distribution $x_0\sim q_0'(x_0)$ and integrating along the original vector field $v_t$ to obtain $x_1$, which then contains the targeted inverse estimate $\hat{K}$. Thus, the samples themselves are not masked during generation — the mask is instead applied to appropriately tilt the source distribution.
>
> > Usefulness of the "validity of inverse estimate"-metric
>
> We argue that a close match between the jointly sampled pressure $H\odot \hat{p}$ and the target pressure $y$ does not necessarily imply an accurate inverse estimate $\hat{K}$. An accurate inverse estimate $\hat{K}$ is characterized by a close match between the corresponding true pressure $H\odot p_{\hat{K}}$ and the target $y$, where $p_{\hat{K}}$ is computed using numerical integration of the PDE with $K=\hat{K}$. To illustrate the disconnect, consider the g-cov methods: here, $H\odot \hat{p}$ closely approximates the target $y$ (Figure 10) while $H\odot p_{\hat{K}}$ does not (Figure 6). The reason for this is that optimizing the guidance cost $\|y-H\odot \hat{p}\|$ may drive the flow to highly unlikely regions in the prior $q_1$. This is manifested by a poor physical consistency between $\hat{K}$ and $\hat{p}$, such that $p_{\hat{K}}$ is very different from $\hat{p}$. Hence, a low guidance cost $\|y-H\odot \hat{p}\|$ here does not imply an accurate inverse estimate, i.e., a low error $\|y-H\odot p_{\hat{K}}\|$. Thus, we argue that $\|y-H\odot p_{\hat{K}}\|$, defined as the "validity of inverse estimate"-metric, is an appropriate choice of evaluation metric.

---

### Official Review · Reviewer_3Ext · 2025-11-11

**Soundness:** 3
**Presentation:** 3
**Contribution:** 2
**Rating:** 6
**Confidence:** 2

**Summary:**

This paper introduces Source-Guided Flow Matching (SGFM), a novel framework that guides generative models by modifying the source distribution rather than the vector field. The key motivation is to keep the pre-trained vector field intact while only adjusting the source distribution to achieve conditional generation. Specifically, this approach transforms the guidance problem into a well-defined task of sampling from a modified source distribution. This work also demonstrates that this approach precisely recovers the desired target distribution under ideal conditions and provides Wasserstein error bounds for cases involving approximate samplers and vector fields.

**Strengths:**

1. The core methodology is grounded in a theoretical result, Theorem 1, which states that if a vector field with flow map transports a source distribution to a target distribution, then for a desired new target distribution, there exists a modified source distribution that is precisely transported to the target one by the *same* vector field. This theorem provides the foundation for SGFM.

2. The framework also provides theoretical guarantees to ensure the quality of generated samples. Besides, the practical implementation of SGFM hinges on accurately and efficiently sampling from the modified source distribution. The paper emphasizes the flexibility to choose sampling methods based on problem specifics.  Experimental evaluations on synthetic 2D datasets demonstrate SGFM-IS's superior sample quality (lower Wasserstein distance) and asymptotic exactness.

**Weaknesses:**

I am not familiar with the field. Here are a few comments:

1. Practical experimental setup: while this work provides solid theoretical derivations guaranteeing the quality of generated samples, all experiments were conducted on relatively small datasets. For large-scale datasets, such as the class-to-image task on ImageNet with vanilla diffusion model, i.e. DiT and SiT, it remains to be seen whether this work can also achieve high-quality generated samples. I hope the authors can provide implementation results on this task, and it is suggested to report gFID and gIS metrics.

2. How to obtain the value of $J$: How should this parameter be handled in practice, given its critical impact on the modified target distribution?

3. The paper clearly notes that optimization-based methods (like D-Flow) can suffer from mode collapse, but does not propose a general solution beyond using more advanced samplers like HMC.

4. The theory assumes access to an optimal or near-optimal vector field. In practice, learning such a field is non-trivial, especially in high dimensions.

5. While the method is general, most experiments use Gaussian source distributions. More analysis on complex source distributions would be beneficial.

**Questions:**

N/A

---

> ### Author Response · Authors · 2025-11-19
>
> Thank you for the helpful review and feedback. We address the weaknesses and questions raised by you below.
>
> >Large-scale dataset: class-to-image task on ImageNet
>
> We thank the reviewer for suggesting this experimental setting. In Section 5.3 we offer experimental evidence for the effectiveness of SGFM in image inverse problems on CelebA, which has a similar scale as ImageNet. Besides, we have evaluated our methods across diverse tasks, including 2D examples, Darcy flow, MNIST and ODE trajectory generation. The total computational usage was approximately 2500 GPU hours. We believe these experiments demonstrate the effectiveness of our approach and are happy to address any specific points the reviewer would like us to clarify.
>
> > How to obtain the values of J?
>
> J is a task-specific loss function reflecting the constraint, such that the likelihood of constraint satisfaction is expressed on the form $e^{-J(x_1)}$. The function can either be designed by hand based on the specific problem or learned using data [R1]. For example, in the Darcy flow experiment, $J$ is the target reconstruction error, i.e., $x_1=[\hat{K}, \hat{p}]$ and $J(x_1)=\|y-H\odot\hat{p}\|$ where $H$ is a binary matrix indicating the observed spatial locations, $\odot$ is the Hadamard product, and $y$ is the observations. Similarly, in the inverse imaging problems, $J(x_1)=\|y-H(x_1)\|^2$ where $H$ applies the degradation and $y$ is the observed (degraded) image. Note that defining $J$ is part of the problem formulation, and is independent of the guidance methodology. Our SGFM framework can accommodate any properly defined $J$, both differentiable and non-differentiable. Therefore, we believe it should not be regarded as a weakness of SGFM.
>
> > Mode collapse
>
> We acknowledge that the efficiency of optimization-based sampling methods comes at the cost of potential mode collapse. To mitigate the issue, we propose to modify the objective term representing the prior probability to a regularizer or to projection (equation 4-6) promoting sample diversity. Generally, SGFM allows the user to balance exactness and efficiency according to their priorities: if the task is to generate a faithful empirical representation of the target distribution, we recommend exact samplers like HMC. If, on the other hand, only a handful of high-probability samples are to be generated with efficiency, we recommend optimization-based sampling methods. Crucially, SGFM flexibly supports both options side-by-side.
>
> > The theory assumes access to an optimal or near-optimal vector field, and learning near-optimal vector field is non-trivial.
>
> We clarify that our theory (Theorems 1 and 2) holds for arbitrary vector fields instead of only near-optimal ones. In other words, SGFM is theoretically applicable for any pre-trained vector fields. The proposal to learn the near-optimal vector field is a practical consideration. On one hand, optimal vector field is less sensitive to approximation errors of the vector field, see Theorem 2 and Appendix C.1.2. On the other hand, optimal vector field acclerates the inference as a small number of function evaluations (NFE) is needed.
>
> We acknowledge that learning the optimal vector fields is challenging, but argue that mini-batch optimal-transport training [R2] allows us to learn vector fields that are _sufficiently_ optimal for effective guidance. In the 2D experiments, performance does not improve for an increasing NFE from 1 to 100 (Figure 4). Similarly, in the CelebA experiment, increasing NFE beyond NFE=3 has little to no added benefit for SGFM-OPT performance (Table 5 in the paper). These results demonstrate that we have successfully learned the near-optimal vector fields, whose guidance performance is satisfactory for our purposes.
>
>
> > Complex source distributions
>
> We would like to highlight additional experiments in the Appendix where the source distribution is not Gaussian. In Figure 7 (Appendix C.1.3), we include three 2D examples with the source distributions being 8-Gaussian, uniform distribution on the unit circle, and uniform distribution within a square, for which SGFM-IS achieves superior results.
>
> While the more complex experiments indeed have Gaussian source distributions, we note that the modified source distribution $q_0'(x_0)\propto q_0(x_0)e^{-J\circ T^*(x_0)}$ generally is  complex due to $J$ even if $q_0$ is simple. For example, the deblurring degragation function $H$ involves convolution. Thus, we expect SGFM to perform similarly even in the case of non-Gaussian source distributions.
>
> [R1] Masatoshi Uehara, et al. Fine-Tuning of Continuous-Time Diffusion Models as Entropy-Regularized Control. 2024.
>
> [R2] Tong A, et al. Improving and generalizing flow-based generative models with minibatch optimal transport. 2023.

---

### Author Response · Authors · 2025-12-03

Dear Area Chair and reviewers,

We sincerely thank you for your time, thoughtful feedback, and engagement with our work. We have incorporated the feedback to revise the manuscript thoroughly, with changes marked in blue in the updated submission. We would like to summarize the discussion and make a few final comments to assist AC in evaluating our paper.

To begin with, we briefly restate the key contributions:
+ **Novel perspective on guidance:** We propose an asymptotically exact, training-free guidance framework for flow matching by modifying the source distribution, supported by rigorous theoretical guarantees and error bound analysis.
+ **Versatile and adaptable framework:** Our framework allows the user to flexibly balance exactness and efficiency through their choice of source distribution sampling method. We illustrate this with samplers ranging from exact methods (IS, HMC) to efficient optimization-based approximations, and we introduce a set of regularizers that promote sample diversity in the latter.
+ **Broad empirical validation:** We demonstrate the framework's flexibility and effectiveness on diverse tasks, including 2D examples, MNIST, ODE trajectory generation, Darcy Flow, and the high-dimensional CelebA dataset. Results demonstrate that our framework consistently achieves state-of-the art performance.

**Reviewer 3Ext**  (rating 6, confidence 2) appreciates the theoretical grounding, flexibility of the framework, and experiments. In response to their concern that learning near-optimal vector fields is challenging, we clarify that our theoretical results do _not_ rely on vector-field optimality; rather, we show that the induced straightness mitigate propagation of approximation errors and accelerates inference. In experiments added to the main text, we show that mini-batch optimal transport training [R1] yields _sufficiently_ optimal vector fields for effective guidance. Regarding their remaining points, we clarify how to design J for different problems, emphasize the proposed regularizers that address the mode collapse problem, and highlight additional experiments in the Appendix that use non-Gaussian source distributions.

**Reviewer T7yj** (rating 6, confidence 2) raises the theoretical analysis, idea, and writing as key strengths. In response to their minor concerns and questions on the Darcy flow experiment, we clarify why our method outperforms the PnP flow baseline, explain how we condition on the partial pressure field, and argue that the "validity of inverse estimate"-metric is an appropriate choice of evaluation metric. We have also updated the paper with clarified notation following their request.

**Reviewer Tm1L** (rating 6, confidence 4) values the flexibility of the framework, the comprehensive experiments, and the writing. In terms of limitations, they request an extended discussion around regularization for non-Gaussian source distributions; in response, we generalize the design procedure and apply to five new types of distributions and add it to the paper. They also highlight the need for a variance interval in Table 2, which we additionally provide. Finally, we address why our framework is more appropriate than the suggested additional baseline [R2] in the setting we consider, i.e., when the constraint is represented by a closed-form cost function J.

**Reviewer wX5E** (rating 4, confidence 3) appreciates our theoretical contribution and comprehensive experiments, while raising concerns about sampling efficiency due to backpropagation through the ODE trajectory. In response, we first emphasize that backpropagation is widely used in training-free guidance, is practically feasible, and can be made even more memory-efficient using advanced techniques such as adjoint methods.
Second, we introduce a novel insight for improving efficiency by exploiting nearly straight vector fields. For example, on CelebA super-resolution tasks, our method reduces the sampling time from 12.5 minutes (using the D-Flow baseline) to just 1.5 minutes.
Regarding their comment on benchmarking against fine-tuning velocity neural network using RL, we believe a fair comparison is difficult since RL is training-based whereas our method is training-free.
Finally, regarding applications to non-differentiable J, we confirmed that our theory is indeed applicable, and in implementation advanced sampling techniques such as elliptical slice sampling can be employed.

In conclusion, our framework is intuitive, comes with rigorous theoretical guarantees, and achieves state-of-the-art performance in all tested tasks. **Although we did not receive any further comments from reviewers after our initial responses, we believe our clarifications and revisions have addressed all of their concerns and strengthens the case for acceptance.**

[R1] Tong A, et al. Improving and generalizing flow-based generative models with minibatch optimal transport. arxiv. 2023.

[R2] Xie, T., et al. Reflected flow matching. 2024.

---

### Meta-Review · Area_Chair_5KZA · 2025-12-26

**Summary:**

This paper introduces a novel method for the conditional sampling problem, which involves sampling from a tilted distribution. The approach reduces this task to tilting the source distribution, for which the authors propose several practical sampling techniques including importance sampling, Monte Carlo, and optimization-based methods.

Reviewer Feedback
Pros: Reviewers praised the theoretically justified approach and the paper's clear writing.
Cons: Primary concerns centered on the method's limited scope: it was validated only on small datasets, operated in a simplified setting, and lacked evidence of scalability to larger datasets. Reviewers also noted its reliance on a “near” optimal transport velocity field, which is simpler to achieve for smaller / lower-dim datasets, leading to potential low applicability in scaled scenarios.

The AC ultimately recommends acceptance, acknowledging the potentially low applicability but favoring the distinctive and novel approach taken for conditioning flows using modification of the source distribution alone.

**Reviewer Concerns:**

please see above.

**Reviewer Scores:**

please see above.

---

### Decision · Program_Chairs · 2026-01-26

Accept (Poster)